

# A multi-model CMIP6-PMIP4 study of Arctic sea ice at 127 ka: sea ice data compilation and model differences

**Masa Kageyama**[1,★], **Louise C. Sime**[2,★], **Marie Sicard**[1,★], **Maria-Vittoria Guarino**[2,★], **Anne de Vernal**[3,★], **Ruediger Stein**[5,6,★], **David Schroeder**[4], **Irene Malmierca-Vallet**[2], **Ayako Abe-Ouchi**[7], **Cecilia Bitz**[8], **Pascale Braconnot**[1], **Esther C. Brady**[9], **Jian Cao**[10], **Matthew A. Chamberlain**[11], **Danny Feltham**[4], **Chuncheng Guo**[12], **Allegra N. LeGrande**[13], **Gerrit Lohmann**[5], **Katrin J. Meissner**[14], **Laurie Menviel**[14], **Polina Morozova**[15], **Kerim H. Nisancioglu**[16,17], **Bette L. Otto-Bliesner**[9], **Ryouta O'ishi**[7], **Silvana Ramos Buarque**[18], **David Salas y Melia**[18], **Sam Sherriff-Tadano**[7], **Julienne Stroeve**[TS1][19], **Xiaoxu Shi**[5], **Bo Sun**[10], **Robert A. Tomas**[9], **Evgeny Volodin**[20], **Nicholas K. H. Yeung**[14], **Qiong Zhang**[21], **Zhongshi Zhang**[TS2][22,11], **Weipeng Zheng**[23], and **Tilo Ziehn**[24]

[1]Laboratoire des Sciences du Climat et de l'Environnement, Institut Pierre Simon Laplace,
Université Paris-Saclay, 91191 Gif-sur-Yvette CEDEX, France
[2]British Antarctic Survey, Cambridge, UK
[3]Geotop, Département des sciences de la Terre et de l'atmosphère, Université du Québec à Montréal, Montréal, Canada[CE1]
[4]Centre for Polar Observation and Modelling, Department of Meteorology, University of Reading, Reading, UK
[5]Alfred Wegener Institute Helmholtz Centre for Polar and Marine Research, Bremerhaven, Germany
[6]MARUM – Center for Marine Environmental Sciences and Faculty of Geosciences,
University of Bremen, Bremen, Germany
[7]Atmosphere and Ocean Research Institute, The University of Tokyo, Tokyo, Japan
[8]Department of Atmospheric Sciences, University of Washington, Seattle, USA
[9]Climate and Global Dynamics Laboratory, National Center for Atmospheric Research, Boulder, USA
[10] Earth System Modeling Center, Nanjing University of Information Science and Technology, Nanjing, 210044, China
[11]CSIRO Oceans and Atmosphere, Hobart, Australia
[12]NORCE Norwegian Research Centre, Bjerknes Centre for Climate Research, Bergen, Norway
[13]NASA Goddard Institute for Space Studies, 2880 Broadway, New York, NY 10025, USA
[14]Climate Change Research Centre, ARC Centre of Excellence for Climate Extremes,
The University of New South Wales, Sydney, Australia
[15]Institute of Geography, Russian Academy of Sciences, Staromonetny L. 29, Moscow, 119017, Russia
[16]Department of Earth Science, University of Bergen, Bjerknes Centre for Climate Research,
Allégaten 41, Bergen, Norway
[17]Centre for Earth Evolution and Dynamics, University of Oslo, Oslo, Norway
[18]Centre National de Recherches Météorologiques, Université de Toulouse, Météo-France,
CNRS (Centre National de la Recherche Scientifique), Toulouse, France
[19]Centre for Earth Observation Science, 535 Wallace Building, University of Manitoba, Winnipeg, MB R3T 2N2 Canada
[20]Marchuk Institute of Numerical Mathematics, Russian Academy of Sciences, ul. Gubkina 8, Moscow, 119333, Russia
[21]Department of Physical Geography, Stockholm University, Stockholm, Sweden
[22]Department of Atmospheric Science, School of Environmental Studies,
China University of Geoscience (Wuhan), Wuhan, China
[23]LASG, Institute of Atmospheric Physics, Chinese Academy of Sciences, Beijing 100029, China
[24]CSIRO Oceans and Atmosphere, Aspendale, Australia
★These authors contributed equally to this work.

**Correspondence:** Masa Kageyama (masa.kageyama@lsce.ipsl.fr)

Published by Copernicus Publications on behalf of the European Geosciences Union.

Received: 22 December 2019 – Discussion started: 23 January 2020
Revised: 14 August 2020 – Accepted: 7 September 2020 – Published:

**Abstract.** The Last Interglacial period (LIG) is a period with increased summer insolation at high northern latitudes, which results in strong changes in the terrestrial and marine cryosphere. Understanding the mechanisms for this response via climate modelling and comparing the models' representation of climate reconstructions is one of the objectives set up by the Paleoclimate Modelling Intercomparison Project for its contribution to the sixth phase of the Coupled Model Intercomparison Project. Here we analyse the results from 16 climate models in terms of Arctic sea ice. The multi-model mean reduction in minimum sea ice area from the pre industrial period (PI) to the LIG reaches 50 % (multi-model mean LIG area is $3.20 \times 10^6$ km$^2$, compared to $6.46 \times 10^6$ km$^2$ for the PI). On the other hand, there is little change for the maximum sea ice area (which is $15$–$16 \times 10^6$ km$^2$ for both the PI and the LIG. To evaluate the model results we synthesise LIG sea ice data from marine cores collected in the Arctic Ocean, Nordic Seas and northern North Atlantic. The reconstructions for the northern North Atlantic show year-round ice-free conditions, and most models yield results in agreement with these reconstructions. Model–data disagreement appear for the sites in the Nordic Seas close to Greenland and at the edge of the Arctic Ocean. The northernmost site with good chronology, for which a sea ice concentration larger than 75 % is reconstructed even in summer, discriminates those models which simulate too little sea ice. However, the remaining models appear to simulate too much sea ice over the two sites south of the northernmost one, for which the reconstructed sea ice cover is seasonal. Hence models either underestimate or overestimate sea ice cover for the LIG, and their bias does not appear to be related to their bias for the pre-industrial period. Drivers for the inter-model differences are different phasing of the up and down short-wave anomalies over the Arctic Ocean, which are associated with differences in model albedo; possible cloud property differences, in terms of optical depth; and LIG ocean circulation changes which occur for some, but not all, LIG simulations. Finally, we note that inter-comparisons between the LIG simulations and simulations for future climate with moderate ($1$ % yr$^{-1}$) CO$_2$ increase show a relationship between LIG sea ice and sea ice simulated under CO$_2$ increase around the years of doubling CO$_2$. The LIG may therefore yield insight into likely 21st century Arctic sea ice changes using these LIG simulations.

## 1 Introduction

The Last Interglacial period (LIG) was the last time global temperature was substantially higher than the pre-industrial period (PI) at high northern latitudes. It is important in helping us understand warm-climate sea ice and climate dynamics (Otto-Bliesner et al., 2013, 2017; Capron et al., 2017; Fischer et al., 2018). Stronger LIG spring and summertime insolation contributed to this warmth, as well as feedbacks amplifying the initial insolation signal, in particular feedbacks related to the marine and land cryosphere. Previous climate model simulations of the LIG, forced by appropriate greenhouse gas (GHG) and orbital changes, have failed to capture the observed high temperatures at higher latitudes (Malmierca-Vallet et al., 2018; Masson-Delmotte et al., 2011; Otto-Bliesner et al., 2013; Lunt et al., 2013). Models used during the previous Coupled Model Intercomparison Project 5 (CMIP5) disagree on the magnitude of Arctic sea ice retreat during the LIG: the diversity of sea ice behaviour across models was linked to the spread in simulated surface temperatures and in the magnitude of the polar amplification (Otto-Bliesner et al., 2013; Lunt et al., 2013; IPCC, 2013). However it was difficult to compare some of the LIG simulations because they were not all run using identical protocol. These studies thus highlighted the need of a systematic approach to study the role of Arctic sea ice changes during the LIG.

Coupled Model Intercomparison Projects (CMIPs) coordinate and design climate model protocols for past, present and future climates and have become an indispensable tool to facilitate our understanding of climate change (IPCC, 2007, 2013; Eyring et al., 2016). The Paleoclimate Model Intercomparison Project 4 (PMIP4) is one of the individual Model Intercomparison Projects that is taking part in CMIP6 (Kageyama et al., 2018). Within this framework, a common experimental protocol for LIG climate simulation was developed by Otto-Bliesner et al. (2017). CMIP models differ among each other in their physical formulation, numerical discretisation and code implementation. However, this CMIP6-PMIP4 LIG standard protocol facilitates model intercomparison work.

Alongside a previous lack of a common experimental protocol, our ability to evaluate CMIP models has previously been hindered by difficulties in determining LIG sea ice extent from marine core evidence (e.g. Otto-Bliesner et al., 2013; Sime et al., 2013; Malmierca-Vallet et al., 2018; Stein et al., 2017). Planktonic foraminifer assemblages that include a subpolar component suggest reduced sea ice in the Arctic Ocean (Nørgaard-Pedersen et al., 2007; Adler et al., 2009). Microfauna found in LIG marine sediments recovered from

the Beaufort Sea Shelf, an area characterised by ice-free conditions during summers today, also support ice-free conditions during those times; this indicates that more saline Atlantic water was present on the Beaufort Shelf, suggesting reduced perennial Arctic sea ice during some part of the LIG (Brigham-Grette and Hopkins, 1995). On the other hand, a reconstruction of LIG Arctic sea ice changes based on sea ice biomarker proxies (see below for details) suggests that the central part of the LIG Arctic Ocean remained covered by ice throughout the year, while a significant reduction of LIG sea ice occurred across the Barents Sea continental margin (Stein et al., 2017). On the modelling side, no previous coupled climate model has simulated an ice-free Arctic during the LIG (Otto-Bliesner et al., 2006; Lunt et al., 2013; Otto-Bliesner et al., 2013; Stein et al., 2017).

Here we address the question of LIG Arctic sea ice by providing a new marine core synthesis. Additionally, the CMIP6-PMIP4 LIG experimental protocol developed by Otto-Bliesner et al. (2017) provides the systematic framework to enable us to examine the question of the simulation of LIG Arctic sea ice using a multi-model approach. This is important given the current level of interest in the ability of climate models to accurately represent key Arctic climate processes during warm periods, including sea ice formation and melting. We compare the LIG Arctic sea ice simulated by each model against our new data synthesis and investigate why different models show different Arctic sea ice behaviour.

## 2   Materials and methods

### 2.1   Current Arctic sea ice

Our main objective is to investigate LIG sea ice. However, a quick assessment of the sea ice simulated in the reference state, i.e. the pre-industrial control experiment (referred to as *piControl* in the CMIP6 terminology, and PI in this paper) was necessary. In the absence of extensive sea ice data for the PI, we used data for a recent period before the current sea ice cover significant decrease. We use the NOAA Optimum Interpolation version 2 data (Reynolds et al., 2002) for the period 1982 to 2001. The sea ice data in this dataset are obtained from different satellite and in situ observations. We have used the monthly time series at a resolution of 1°. This dataset is termed "NOAA_OI_v2" in the rest of this paper.

### 2.2   Marine records of LIG Arctic sea ice

We focus here on records of sea ice from marine cores. Table 1 provides a summary of LIG sea ice information and data obtained from marine sediment cores collected in the Arctic Ocean, Nordic Seas and northern North Atlantic. South of 78° N, the records show ice-free conditions. Most of these sea ice records are derived from quantitative estimates of sea surface parameters based on dinoflagellate cysts (dinocysts). North of 78° N the sea-ice-related records are rare and dif-

ferent types of indicators were used. In addition to dinocysts, the records are based on biomarkers linked to phototrophic productivity in sea ice and on foraminifers and ostracods that both provide indication on water properties and indirectly on sea ice (de Vernal et al., 2013b). Between 78 and 87° N, the faunal data have been interpreted as indicating seasonal sea ice cover conditions during the LIG.

Among sea ice cover indicators, dinocyst assemblages have been used as quantitative proxy based on the application of the modern analogue technique applied to a standardised reference modern data base developed from surface sediment samples collected at middle to high latitudes of the Northern Hemisphere (de Vernal et al., 2005, b, 2013b, 2020). The sea ice estimates from dinocysts used here are from different studies (see references in Table 1) and reconstructions based the new database, including 71 taxa and 1968 stations (de Vernal et al., 2020). The reference sea ice data used for calibration are the monthly 1955–2012 average of the National Snow and Ice Data Center (NSIDC) (Walsh et al., 2016). The results are expressed in term of annual mean of sea ice cover concentration or as the number of months with $> 50\%$ of sea ice. The error of prediction for sea ice concentration is $\pm 12\%$ and that of sea ice cover duration through the year is $\pm 1.5$ months $yr^{-1}$. Such values are very close to the interannual variability in areas occupied by seasonal sea ice cover (see de Vernal et al., 2013b).

Our biomarker approach for sea ice reconstruction is based on the determination of a highly branched isoprenoid (HBI) with 25 carbons (C25 HBI monoene = IP25) (Belt et al., 2007). This biomarker is only biosynthesised by specific diatoms living in the Arctic sea ice (Brown et al., 2014), meaning the presence of IP25 in the sediments is a direct proof for the presence of past Arctic sea ice. Meanwhile, this biomarker approach has been used successfully in numerous studies dealing with the reconstruction of past Arctic sea ice conditions during the late Miocene to Holocene (for a review, see Belt, 2018). By combining the sea ice proxy IP25 with (biomarker) proxies for open-water (phytoplankton productivity such as brassicasterol, dinosterol or a specific tri-unsaturated HBI, HBI-III), the so-called PIP25 index has been developed (Müller et al., 2011; Belt et al., 2015; Smik et al., 2016). Based on a comparison ("calibration") PIP25 data obtained from surface sediments with modern satellite-derived (spring) sea ice concentration maps (Müller et al., 2011; Xiao et al., 2015; Smik et al., 2016), the PIP25 approach may allow a more semi-quantitative reconstruction of present and past Arctic Ocean sea ice conditions from marine sediments, i.e. estimates of spring sea ice concentration (or in the Central Arctic probably more the summer situation due to light limitations for algae growth in the other seasons). Based on these data, one may separate "permanent to extended sea ice cover" ($> 0.75$) and "seasonal sea ice cover"; (0.75–0.1), perhaps including the sub-groups "ice-edge" (0.75–0.5) and "less/reduced sea ice" (0.5–0.1), and "ice-free" ($< 0.1$). For

https://doi.org/10.5194/cp-16-1-2020

**Table 1.** Marine core records of Arctic sea ice from MIS5e. The references indicated for the dinocyst reconstructions are those for the initial core, the reconstruction itself follows de Vernal et al. (2013a, b, 2020) (see the main text for details). SIC stands for sea ice concentration. Question marks indicate that seasonal duration or annual mean SIC are uncertain and not quantitatively estimated. More general qualitative statements are still possible and are given in the "Qualitative sea ice state" column.

| Latitude (°N) | Longitude (°E) | Sea ice indicator | Core name | Reference | Site no. on map | Chronol. control 1 = good 2 = uncertain | Qualitative sea ice state | Duration of SIC > 0.50, in months per year Min | Max | Annual mean SIC Min | Max |
|---|---|---|---|---|---|---|---|---|---|---|---|
| 87.08 | 144.77 | Ostracod fauna | Oden96/12-1pc | Cronin et al. (2010) | 6 | 2 | Perennial sea ice (summer sea ice concentration > 75 %) | ? | ? | ? | ? |
| 85.32 | −14 | IP25/PIP25 | PS2200-5 | Stein et al. (2017) | 8 | 2 | Perennial sea ice | ? | ? | ? | ? |
| 85.32 | −14 | Ostracod fauna | PS2200-5 | Cronin et al. (2010) | 8 | 2 | Perennial sea ice (summer sea ice concentration > 75 %) | ? | ? | ? | ? |
| 85.14 | −171.43 | IP25/PIP25 | PS51/38-3 | Stein et al. (2017) | 5 | 2 | Perennial sea ice | ? | ? | ? | ? |
| 84.81 | −74.26 | Subpolar foraminifers | GreenICE (core 11) | Nørgaard-Pedersen et al. (2007) | 7 | 2 | Reduced sea ice cover, even partly seasonally ice-free (but with regional signal or just local polynya conditions) | ? | ? | ? | ? |
| 81.92 | 13.83 | IP25/PIP25 | PS92/039-2 | Kremer et al. (2018b) | 10 | 1 | Perennial sea ice (summer sea ice concentration > 75 %) | ? | ? | ? | ? |
| 81.54 | 30.17 | Dinocysts | PS2138-1 | Matthiessen et al. (2001), MatthiessenKnies_2001 [TS3] | 9 | 1 | Seasonal sea ice conditions summer probably ice-free | 0 | 5 | 0 | 0.3 |
| 81.54 | 30.59 | IP25/PIP25 | PS2138-1 | Stein et al. (2017) | 9 | 1 | Seasonal sea ice conditions (summer probably ice-free) | ? | ? | 0.1 | 0.3 |
| 81.19 | 140.04 | IP25/PIP25 | PS2757-8 | Stein et al. (2017) | 4 | 2 | Perennial sea ice | ? | ? | ? | ? |
| 79.59 | −172.50 | Subpolar foraminifers | HLY0503-8JPC | Adler et al. (2009) | 3 | 2 | Seasonal sea ice conditions (summer probably ice-free) | ? | ? | ? | ? |
| 79.32 | −178.07 | Ostracod fauna | NP26-32 | Cronin et al. (2010) | 1 | 2 | Perennial sea ice (summer sea ice concentration > 75 %) | ? | ? | ? | ? |

| Latitude (°N) | Longitude (°E) | Sea ice indicator | Core name | Reference | Site no. on map | Chronol. control 1 = good 2 = uncertain | Qualitative sea ice state | Duration of SIC > 0.50, in months per year Min | Max | Annual mean SIC Min | Max |
|---|---|---|---|---|---|---|---|---|---|---|---|
| 79.20 | 4.67 | IP25/PIP25 | PS93/006-1 | Kremer et al. (2018a) | 11 | 1 | Seasonal sea ice conditions (summer probably ice-free) | ? | ? | 0.3 | 0.6 |
| 78.98 | −178.15 | Ostracod fauna | NP26-5 | Cronin et al. (2010) | 2 | 2 | Perennial sea ice (summer sea ice concentration > 75 %) | ? | ? | ? | ? |
| 76.85 | 8.36 | Dinocysts | M23455-3 | Van Nieuwenhove et al. (2011) | 12 | 1 | Nearly ice-free all year round | 0 | 1 | 0 | 0.15 |
| 70.01 | −12.43 | Dinocysts | M23352 | Van Nieuwenhove et al. (2013) | 13 | 1 | Nearly ice-free all year round | 0 | 1 | 0 | 0.15 |
| 69.49 | −17.12 | Dinocysts | PS1247 | Nicolas Van Nieuwenhove (personal communication, 2019), chronology from Zhuravleva et al. (2017) | 14 | 1 | Nearly ice-free all year round | 0 | 2 | 0 | 0.2 |
| 67.77 | 5.92 | Dinocysts | M23323 | Van Nieuwenhove et al. (2011) | 15 | 1 | Nearly ice-free all year round | 0 | 1 | 0 | 0.15 |
| 67.09 | 2.91 | Dinocysts | M23071 | Van Nieuwenhove et al. (2008); Van Nieuwenhove and Bauch (2008) | 16 | 1 | Nearly ice-free all year round | 0 | 1 | 0 | 0.15 |
| 60.58 | −22.07 | Dinocysts | MD95-2014 | Eynaud (1999) | 17 | 1 | Ice-free all year round | 0 | 0 | 0 | 0 |
| 58.77 | −25.95 | Dinocysts | MD95-2015 | Eynaud et al. (2004) | 18 | 1 | Ice-free all year round | 0 | 0 | 0 | 0 |
| 58.21 | −48.37 | Dinocysts | HU90-013-13P | Hillaire-Marcel et al. (2001), de Vernal and Hillaire-Marcel (2008) | 19 | 1 | Nearly ice-free all year round | 0 | 1 | 0 | 0.15 |
| 55.47 | −14.67 | Dinocysts | MD95-2004 | Van Nieuwenhove et al. (2011) | 20 | 1 | Ice-free all year round | 0 | 0 | 0 | 0 |

| Latitude (°N) | Longitude (°E) | Sea ice indicator | Core name | Reference | Site no. on map | Chronol. control 1 = good 2 = uncertain | Qualitative sea ice state | Duration of SIC > 0.50, in months per year Min | Max | Annual mean SIC Min | Max |
|---|---|---|---|---|---|---|---|---|---|---|---|
| 53.33 | −45.26 | Dinocysts | HU91-045-91 | This paper | 21 | 1 | Ice-free all year round | 0 | 1 | 0 | 0.15 |
| 53.06 | −33.53 | Dinocysts | IODP1304 | This paper, Hodell et al. (2009) | 22 | 1 | Nearly ice-free all year round | 0 | 1 | 0 | 0.15 |
| 50.17 | −45.63 | Dinocysts | IODP1302/1303 | This paper, Hillaire-Marcel et al. (2011) for the chronology | 23 | 1 | Nearly ice-free all year round | 0 | 1 | 0 | 0.15 |
| 46.83 | −9.52 | Dinocysts | MD03-2692 | Penaud et al. (2008) | 24 | 1 | Ice-free all year round | 0 | 0 | 0 | 0.15 |
| 37.80 | −10.17 | Dinocysts | MD95-2042 | Eynaud et al. (2000) | 25 | 1 | Ice-free all year round | 0 | 0 | 0 | 0 |

pros and cons of this approach, we refer to a recent review by Belt (2018).

Based on several IP25/PIP25 records obtained from central Arctic Ocean sediment cores (see Fig. 1 for core locations and Table 1 for data), perennial sea ice cover probably existed during the LIG in the Central Arctic, whereas along the Barents Sea continental margin, influenced by the inflow of warm Atlantic Water, sea ice was significantly reduced (Stein et al., 2017). However, Stein et al. (2017) emphasise that the PIP25 records obtained from the central Arctic Ocean cores indicating a perennial sea ice cover have to be interpreted cautiously as the biomarker concentrations are very low to absent (see Belt, 2018 for further discussion). The productivity of algal material (ice and open water) must have been quite low, so that (almost) nothing reached the seafloor or is preserved in the sediments, and there must have been periods during the LIG when some open-water conditions occurred, since subpolar foraminifers and coccoliths were found in core PS51/038 and PS2200 (Stein et al., 2017). It is however unclear whether these periods equate to more than 1 month $yr^{-1}$ of open water (or seasonal ice conditions). This explains why some sites show both seasonal and perennial interpretations at the same site. The reader is referred to the original publications (Table 1) for more information on these data. Furthermore and importantly, a new revised $^{230}$Th chronology of late Quaternary sequences from the central Arctic Ocean (Hillaire-Marcel et al., 2017) questions the age model of some of the data listed in Table 1. Thus, further verification of age control is still needed and the data from the central Arctic Ocean should be interpreted with caution. We have therefore marked the chronological control as "uncertain" for these cores, while the chronological control is good for cores outside the Central Arctic.

The information given by the different types of sea ice indicators shows that care should be taken when comparing them with model results. We have used the qualitative information given in Table 1, taking into account the threshold given in this table. Indeed, for instance, "perennial sea ice cover" does not automatically mean 100 % sea ice cover, or a sea ice concentration (SIC) of 1.0. It means rather that there is sea ice but not necessarily at a concentration of 100 % over the core site throughout the year (i.e. the summer season is not totally ice-free). Most qualitative reconstructions cite a threshold of 75 %, which we have therefore used in our model-data comparison. We have also used the quantitative mean annual sea ice reconstructions. Finally, similar to studies for future climate, we have considered the Arctic to be ice-free when, in any given month, the total area of sea ice is less than $1 \times 10^6$ km$^2$. This means that some marine core sites could remain ice covered for the summer, but the Arctic would nevertheless remain technically ice-free.

## 2.3 CMIP6-PMIP4 models

The last Coupled Model Intercomparison Project Phase 5 (CMIP5) collected climate simulations performed with 60 different numerical models by 26 research institutes around the world (IPCC, 2013). The follow-on CMIP6 archive, to be completed in 2020, is expected to gather model outputs from over 30 research institutes. Of these, currently 15 models have run the CMIP6-PMIP4 LIG simulation (Table 2). We present results here from all these models.

Table 2 provides an overview of the models used in this study. They are state-of-the-art coupled general circulation models (GCMs) and Earth System Models (ESMs) simulating the atmosphere, ocean, sea ice and land surface processes dynamics with varying degrees of complexity. These 15 CMIP6-PMIP4 models have been developed for several years by individual institutes across the world and, in the context of CMIP6, are used in the same configuration to seamlessly simulate past, present and future climate. We have added the results from the LOVECLIM Earth System Model of Intermediate Complexity, which can be used for longer simulations.

Table 2 shows the following qualities for each model: model denomination, physical core components, horizontal and vertical grid specifications, details on prescribed vs. interactive boundary conditions, relative publication for an in-depth model description, and LIG simulation length (spin-up and production runs).

## 2.4 PMIP4 LIG (*lig127k*) simulation protocol

Results shown here are from the main Tier 1 LIG simulation, from the standard CMIP6-PMIP4 LIG experimental protocol (Otto-Bliesner et al., 2017). The prescribed LIG (*lig127k*) protocol differs from the CMIP6 pre-industrial (PI) simulation protocol in astronomical parameters and the atmospheric trace greenhouse gas concentrations (GHG). LIG astronomical parameters are prescribed according to Berger and Loutre (1991), and atmospheric trace GHG concentrations are based on ice core measurements. Table 3 from Otto-Bliesner et al. (2017) summarises the protocol. All models followed this protocol, except CNRM-CM6-1 for which the most important forcings for the LIG, i.e. the astronomical parameters, have been imposed at the recommended values, but the GHG have been kept at their pre-industrial values of 284.3170 ppm for $CO_2$, 808.2490 ppb for $CH_4$ and 273.0211 ppb for $N_2O$. All other boundary conditions, including solar activity, ice sheets, aerosol emissions, etc., are identical to PI protocol. Both the Greenland and Antarctica ice sheets are known to have shrunk during the interglacial with different timings, and therefore taking PI characteristics for the *lig127k protocol* is an approximation, particularly for the Antarctic ice sheet, which was possibly smaller than in the PI at that time (Otto-Bliesner et al., 2017). The Greenland ice sheet likely reached a minimum at around 120 ka and was probably still

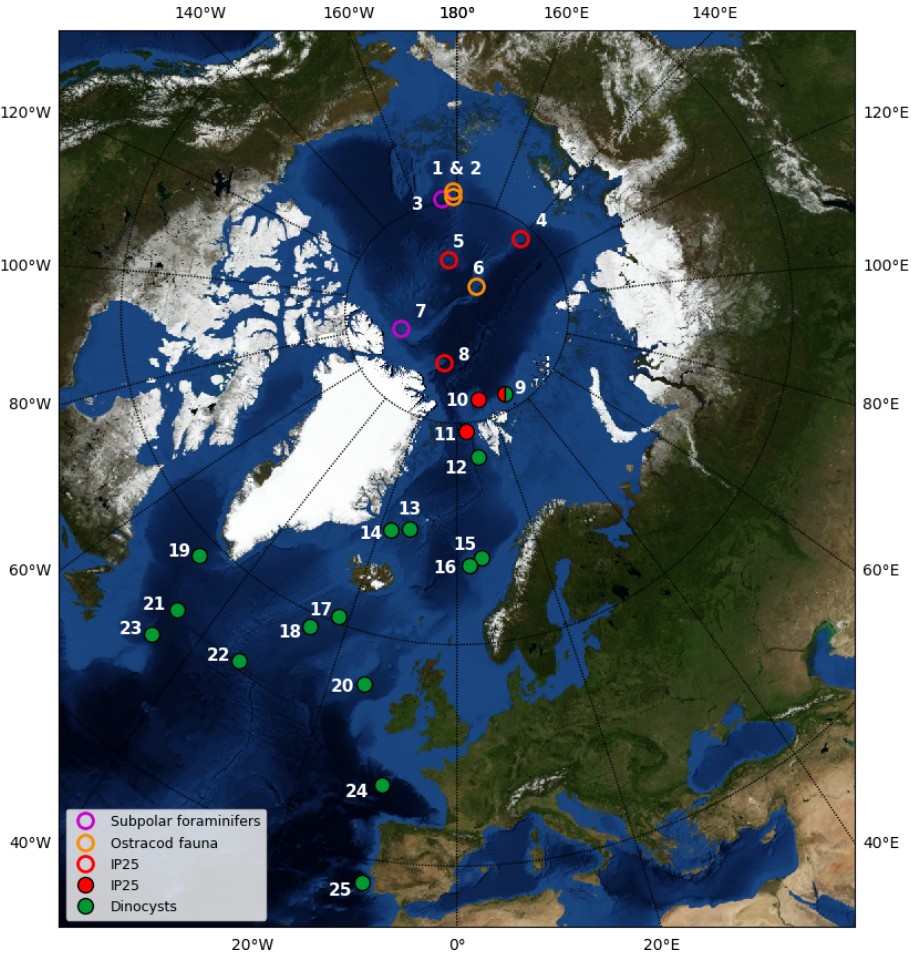

**Figure 1.** Map showing the location of the LIG Arctic sediment cores listed in Table 1. Open symbols correspond to records with uncertain chronology, and filled symbols correspond to records with good chronology. The map background has been created using http://visibleearth.nasa.gov (last access: 1 January 2020).

close to its PI size at 127 ka. Given the dating uncertainties and the difficulty for models to include the largest changes in ice sheets for 127 ka, i.e. changes in West Antarctica, the choice of the PMIIP4 working group on interglacials was to use the PI ice sheets as boundary conditions for the Tier 1 PMIP4-CMIP6 experiments presented here and to foster sensitivity experiments to ice sheet characteristics at a later stage. In terms of the Greenland ice sheet, the approximation is considered quite good and ideal for starting transient experiments through the whole interglacial period.

LIG simulations were initialised either from a previous LIG run, or from the standard CMIP6 protocol pre-industrial simulations, using constant 1850 GHGs, ozone, solar, tropospheric aerosol, stratospheric volcanic aerosol and land use forcing.

Although PI and LIG spin-ups vary between the models, most model groups aimed to allow the land and oceanic masses to attain approximate steady state i.e. to reach atmospheric equilibrium and to achieve an upper-oceanic equilibrium. LIG production runs are all between 100–200 years long, which is generally within the appropriate length for Arctic sea ice analysis (Guarino et al., 2020).

The LIG orbital parameters result in modifications of the definitions of the months and seasons (in terms of start and end dates within a year). Since daily data was not available for all models to re-compute LIG-specific monthly averages, we have corrected these averages using the method of Bartlein and Shafer (2019). Unless otherwise specified, we use these results adjusted for the LIG calendar throughout this paper.

## 2.5 The CMIP6 1pctCO2 protocol

We compare the response to the lig127k forcings to idealised forcings for future climate. We have chosen to use the *1pctCO2* simulation from the CMIP6 DECK (Diagnostic, Evaluation and Characterization of Klima Eyring et al., 2016). These simulations start from the PI (*piControl*) experiment and the atmospheric $CO_2$ concentration is gradually

**Table 2.** Overview of models that have run the CMIP6-PMIP4 LIG simulation. For each model, denomination, physical core components, horizontal and vertical grid specifications, details on prescribed vs. interactive boundary conditions, reference publication, and LIG simulation length are shown. TS4 TS5

| Model name (abbreviation) | Physical core components | Model grid (i_lon × i_lat × z_lev) | Boundary conditions | Reference publication | LIG simulation length (years) |
|---|---|---|---|---|---|
| ACCESS-ESM1-5 (ACCESS) | Atmosphere: UM<br>Land: CABLE2.4<br>Ocean: MOM5<br>Sea Ice: CICE4.1 | Atmosphere:<br>192 × 145 × L38<br>Ocean: 360 × 300 × L50 | Vegetation: prescribed<br>Aerosol: prescribed<br>Ice-Sheet: prescribed | Ziehn et al. (2017) | Spin-up: 400<br>Production: 200 |
| AWIESM-1-1-LR (AWIESM1) | Atmosphere: ECHAM6.3.04p1<br>Land: JSBACH_3.20<br>Ocean: FESOM 1.4<br>Sea Ice: FESOM 1.4 | Atmosphere:<br>192 × 96 × L47<br>Ocean: unstructured grid<br>126 859 nodes × L46 | Vegetation: Interactive<br>Aerosol: prescribed PI<br>Ice-Sheet: prescribed | Sidorenko et al. (2015) | Spin-up: 1000<br>Production: 100 |
| AWIESM-2-1-LR (AWIESM2) | Atmosphere: ECHAM6.3.04p1<br>Land: JSBACH 3.20<br>Ocean: FESOM 2<br>Sea Ice: FESOM 2 | Atmosphere:<br>192 × 96 × L47<br>Ocean: unstructured grid<br>126 858 nodes × L48 | Vegetation: interactive<br>Aerosol: prescribed<br>Ice-Sheet: prescribed | Sidorenko et al. (2019, 2015) | Spin-up: 1000<br>Production: 100 |
| CESM2 | Atmosphere: CAM6<br>Land: CLM5<br>Ocean: POP2<br>Sea Ice: CICE5.1 | Atmosphere:<br>288 × 192 × L32<br>Ocean: 320 × 384 × L60 | Vegetation: prescribed<br>Aerosol: interactive<br>Ice-Sheet: prescribed | Danabasoglu et al. (2020) | Spin-up: 325<br>Production: 700 |
| CNRM-CM6-1 (CNRM-CM6) | Atmosphere: ARPEGE-Climat<br>Land: ISBA-CTRIP<br>Ocean: NEMO3.6<br>Sea Ice: GELATO6 | Atmosphere:<br>256 × 128 × L91<br>(Triangular-Linear 127)<br>Ocean: 362 × 294 × L75 | Vegetation: prescribed<br>Aerosol: prescribed PI<br>Ice-Sheet: prescribed | Voldoire et al. (2019) | Spin-up: 100<br>Production: 200 |
| EC-Earth3 (EC-Earth) | Atmosphere: IFS-cy36r4<br>Land: HTESSEL<br>Ocean: NEMO3.6<br>Sea Ice: LIM3 | Atmosphere:<br>T159(480 × 240) × L62<br>Ocean: 362 × 292 × L75 | Vegetation: prescribed<br>Aerosol: prescribed<br>Ice-Sheet: prescribed | Hazeleger et al. (2012) | Spin-up: 300<br>Production: 200 |
| GISS-E2.1-G | Atmosphere: GISS-E2.1<br>Land: GISSE2.1<br>Ocean & Sea Ice: GISS Ocean v1 | Atmosphere:<br>2 × 2.5 × 40L<br>Ocean: 1 × 1.25 × 40L | Vegetation: Ent/Not Dynamic<br>Aerosol: NINT<br>Ice-Sheet: N/A | Kelley et al. (2020) | Spin-up: 1000<br>Production: 100 |
| HadGEM3-GC3.1-LL (HadGEM3) | Atmosphere: MetUM-GA7.1<br>Land: JULES-GA7.1<br>Ocean: NEMO-GO6.0<br>Sea Ice: CICE-GSI8 | Atmosphere:<br>192 × 144 × L85<br>Ocean: 360 × 330 × L75 | Vegetation: prescribed<br>Aerosol: Prescribed<br>Ice-Sheet: prescribed | Williams et al. (2018) | Spin-up: 350<br>Production: 200 |
| INM-CM4-8 | Atmosphere: INM-AM4-8<br>Land: INM-LND1<br>Ocean: INM-OM5<br>Sea Ice: INM-ICE1 | Atmosphere:<br>180 × 120 × L21<br>Ocean: 360 × 318 × L40 | Vegetation: prescribed<br>Aerosol: interactive<br>Ice-Sheet: prescribed | Volodin et al. (2018) | Spin-up: 50<br>Production: 100 |
| IPSL-CM6A-LR | Atmosphere: LMDZ6<br>Land: ORCHIDEE<br>Ocean: NEMO-OPA<br>Sea Ice: NEMO-LIM3 | Atmosphere:<br>144 × 143 × L79<br>Ocean: 362 × 332 × L75 | Vegetation: prescribed PFTs, interactive phenology<br>Aerosol: Prescribed PI values<br>Ice Sheet: prescribed | Boucher et al. (2019) | Spin-up: 300<br>Production: 200 |

| Model name (abbreviation) | Physical core components | Model grid (i_lon × i_lat × z_lev) | Boundary conditions | Reference publication | LIG simulation length (years) |
|---|---|---|---|---|---|
| LOVECLIM1.2 | Atmosphere: ECBilt Land: VECODE Ocean & Sea Ice: CLIO | Atmosphere: 64 × 32 × L3 Ocean: 120 × 65 × L20 | Vegetation: interactive Aerosol: – Ice Sheet: prescribed | Goosse et al. (2010) | Spin-up: 3000 Production: 1000 |
| MIROC-ES2L | Atmosphere: CCSR AGCM Land: MATSIRO6.0 + VISIT-e Ocean: COCO4.9 Sea Ice: COCO4.9 | Atmosphere: 128 × 64 × L40 Ocean: 360 × 256 × L63 | Vegetation: prescribed Aerosol: prescribed Ice Sheet: prescribed | Hajima et al. (2019), Tatebe et al. (2018) | Spin-up: 1450 Production: 100 |
| NESM3 | Atmosphere: ECHAM6.3 Land: JS-BACH Ocean: NEMO3.4 Sea Ice: CICE4.1 | Atmosphere: 192 × 96 × L47 Ocean: 384 × 362 × L46 | Vegetation: interactive Aerosol: prescribed Ice-Sheet: prescribed | Cao et al. (2018) | Spin-up: 500 Production:100 |
| NorESM1-F (NORESM1) | Atmosphere: CAM4 Land: CLM4 Ocean: MICOM Sea Ice: CICE4 | Atmosphere: 144 × 96 × L26 Ocean: 360 × 384 × L53 | Vegetation: prescribed, as PI Aerosol: prescribed, as PI Ice Sheet: prescribed, as PI | Guo et al. (2019) | Spin-up: 500 Production: 200 |
| NorESM2-LM (NORESM2) | Atmosphere: CAM-OSLO Land: CLM Ocean: BLOM Sea Ice: CICE | Atmosphere: 144 × 96 × L32 Ocean: 360 × 384 × L53 | Vegetation: as in PI Aerosol: as in PI Ice sheet: as in PI | Seland et al. (2019) | Spin-up: 300 Production: 200 |

**Table 3.** Astronomical parameters and atmospheric trace gas concentrations used to force LIG and PI simulations.

| Astronomical parameters | LIG | PI |
|---|---|---|
| Eccentricity | 0.039378 | 0.016764 |
| Obliquity | 24.040° | 23.459° |
| Perihelion-180° | 275.41° | 100.33° |
| Date of vernal equinox | 21 March at noon | 21 March at noon |
| Trace gases | | |
| $CO_2$ | 275 ppm | 284.3 ppm |
| $CH_4$ | 685 ppb | 808.2 ppb |
| $N_2O$ | 255 ppb | 273 ppb |

increased by $1\% \, \mathrm{yr}^{-1}$ for at least 150 years, i.e. 10 years after atmospheric $CO_2$ quadrupling.

# 3 Results: simulated Arctic sea ice

Since all LIG production runs are at least 100 years in length, all model results are averaged over at least 100 years. We refer to the multi-model mean throughout as the MMM. We consider both the sea ice area (SIA), defined as the sum, over all Northern Hemisphere ocean cells, of the sea ice concentration × the cell area and the sea ice extent (SIE), defined as the sum of the areas of ocean cells where the sea ice concentration is larger than 0.15. Both quantities are used in sea ice studies, SIE has been used widely in IPCC AR5 (Vaughan et al., 2013), while SIA tends to be used more for CMIP6 analyses (e.g. SIMIP Community, 2020).

## 3.1 PI sea ice

For the present-day we have satellite and in situ observations with which to evaluate the models. The use of present-day sea ice data implies that we might expect the simulated PI sea ice to be generally somewhat larger than the observed mean. Indeed the atmospheric $CO_2$ levels for the years for which we chose the observation dataset (1982 to 2001) were between 340 and 370 ppm, compared to the PI level of 280 ppm. Figure 2 shows the mean seasonal cycle of the Arctic sea ice extent simulated for the PI and LIG alongside the observed Arctic sea ice extent.

The summer minimum monthly MMM SIA for the PI is $6.46 \pm 1.41 \times 10^6 \, \mathrm{km}^2$, compared to the observed 1981 to 2002 mean of $5.65 \times 10^6 \, \mathrm{km}^2$. In terms of SIE, the summer minimum for PI is $8.89 \pm 1.41 \times 10^6 \, \mathrm{km}^2$, to be compared to the observed $7.73 \times 10^6 \, \mathrm{km}^2$. Interestingly this MMM PI

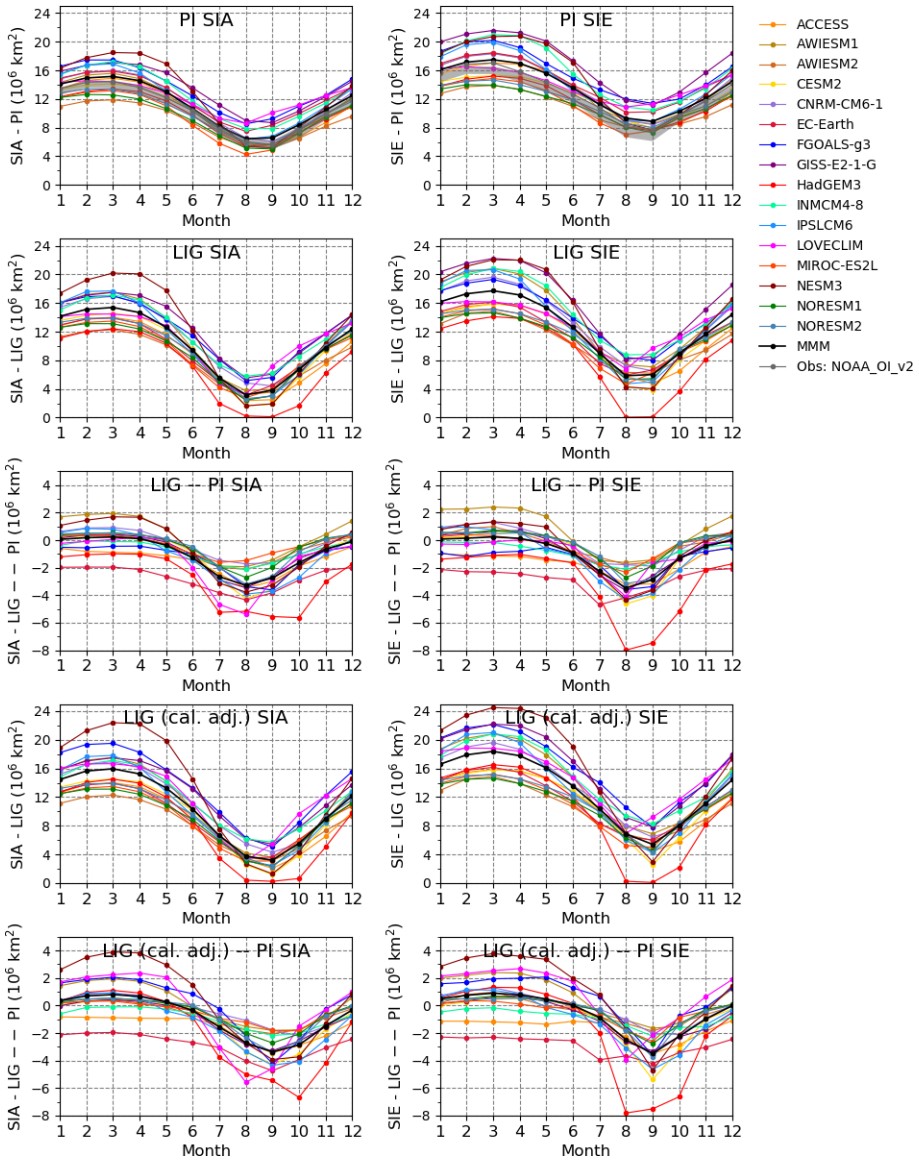

**Figure 2.** Mean seasonal cycle of the Arctic sea ice area (SIA, left-hand side) and sea ice extent (SIE, right-hand side), in $10^6$ km$^2$, simulated for the PI and LIG periods by the PMIP4 models. The top row shows the results for PI. The grey shading shows the monthly minimum and maximum in the SIA and SIE observed over the years 1982–2001, as given by the NOAA_OI_v2 dataset. The second and fourth row show the LIG results, with no calendar adjustment and with calendar adjustment, respectively. The third and bottom row show the corresponding LIG–PI anomalies, with no calendar adjustment and with calendar adjustment, respectively.

area and extent is a little larger than the 1981–2002 area. The majority of the simulations show a realistic representation of the geographical extent for the summer minimum (Fig. 3, Table 4), with 9 out of 16 models showing a slightly smaller area compared to the present-day observations and 7 showing an overestimated area. LOVECLIM, EC-Earth, FGOALS-g3, GISS-E2-1-G and INM-CM4-8 clearly simulate too much ice (Table 4). The other models generally exhibit realistic PI summer minimum ice conditions. The detail of the geographical distribution of sea ice for the models, the MMM and the NOAA_OI_v2 datasets (Fig. 3) con-

firms the results in terms of Arctic sea ice extent. Overestimations appear to be due to too much sea ice being simulated in the Barents–Kara area (LOVECLIM, FGOALS-g3, GISS-E2-1-G), in the Nordic Seas (EC-Earth, FGOALS-g3) and in Baffin Bay (LOVECLIM, INM-CM4-8, EX-Earth). MIROC-ES2L performs rather poorly for the PI, with insufficient ice close to the continents. The other models generally match the 0.15 isoline from the NOAA_OI_v2 dataset in a realistic manner. The winter maximum monthly MMM areas show little difference between the present-day and PI simulated areas. The MMM PI area is $15.16 \pm 1.90 \times 10^6$ km$^2$, com-

**Table 4.** Sea ice area and extent (in $10^6$ km$^2$) for the PI and LIG simulations (calendar-adjusted values). MMM stands for the multi-model mean, SD for the multi-model standard deviation. TS6 TS7

| Model or dataset | PI sea ice area minimum (month) | maximum (month) | LIG sea ice area minimum (month) | maximum (month) | PI sea ice extent minimum (month) | maximum (month) | LIG sea ice extent minimum (month) | maximum (month) |
|---|---|---|---|---|---|---|---|---|
| NOAA_OI_v2 | 5.65 (8) | 14.44 (2) | na | na | 7.73 (8) | 17.05 (2) | na | na |
| ACCESS | 5.49 (8) | 14.90 (2) | 2.05 (8) | 14.01 (2) | 7.93 (8) | 17.04 (2) | 4.44 (8) | 15.85 (2) |
| AWIESM1 | 5.39 (8) | 15.59 (2) | 3.58 (8) | 17.53 (2) | 8.52 (8) | 18.42 (2) | 6.88 (8) | 20.82 (2) |
| AWIESM2 | 5.19 (8) | 11.89 (2) | 3.14 (8) | 12.28 (2) | 7.78 (8) | 13.90 (2) | 5.92 (8) | 14.87 (2) |
| CESM2 | 5.45 (8) | 14.12 (2) | 1.18 (8) | 14.53 (2) | 7.92 (8) | 15.26 (2) | 2.55 (8) | 15.81 (2) |
| CNRM-CM6-1 | 6.07 (8) | 16.02 (2) | 4.29 (8) | 16.94 (2) | 8.44 (8) | 18.32 (2) | 6.41 (8) | 19.62 (2) |
| EC-Earth | 7.49 (7) | 15.89 (2) | 3.46 (7) | 13.93 (2) | 10.13 (7) | 18.46 (2) | 6.01 (8) | 16.16 (2) |
| FGOALS-g3 | 8.54 (7) | 17.46 (1) | 5.04 (8) | 19.51 (2) | 11.40 (7) | 20.20 (1) | 7.78 (8) | 22.14 (2) |
| GISS-E2-1-G | 8.70 (8) | 17.08 (2) | 5.41 (8) | 17.49 (2) | 11.13 (8) | 21.58 (2) | 7.83 (8) | 22.20 (2) |
| HadGEM3 | 5.40 (7) | 13.40 (2) | 0.23 (8) | 14.50 (2) | 7.58 (7) | 15.20 (2) | 0.07 (8) | 16.52 (2) |
| INMCM4-8 | 7.88 (7) | 17.24 (2) | 5.71 (8) | 17.14 (2) | 10.47 (7) | 20.99 (2) | 8.24 (8) | 20.83 (2) |
| IPSLCM6 | 6.39 (7) | 16.91 (2) | 2.46 (8) | 17.82 (2) | 8.88 (7) | 19.91 (2) | 4.24 (8) | 21.02 (2) |
| LOVECLIM | 8.64 (7) | 14.56 (1) | 3.06 (7) | 16.66 (2) | 10.90 (7) | 16.52 (1) | 6.96 (7) | 18.85 (1) |
| MIROC-ES2L | 4.27 (7) | 13.17 (2) | 3.05 (7) | 13.49 (2) | 7.04 (7) | 14.87 (2) | 4.98 (8) | 15.19 (2) |
| NESM3 | 5.20 (8) | 18.50 (2) | 1.28 (8) | 22.39 (2) | 7.67 (8) | 20.80 (2) | 2.96 (8) | 24.50 (2) |
| NORESM1 | 5.03 (8) | 12.64 (1) | 2.31 (8) | 13.11 (2) | 7.30 (8) | 14.00 (1) | 4.52 (8) | 14.62 (2) |
| NORESM2 | 5.62 (8) | 13.38 (1) | 2.22 (8) | 13.89 (2) | 8.02 (8) | 14.66 (1) | 4.26 (8) | 15.12 (2) |
| MMM | 6.46 (7) | 15.16 (2) | 3.20 (8) | 15.95 (2) | 8.89 (7) | 17.48 (2) | 5.39 (8) | 18.38 (2) |
| SD | 1.41 | 1.90 | 1.50 | 2.61 | 1.41 | 1.90 | 2.13 | 3.12 |

pared to the observed 1981 to 2002 mean of $14.44 \times 10^6$ km$^2$. For both the summer and winter, the simulations and observations mostly agree on the month that the minimum and maximum are attained: July–August for the minimum and January–February for the maximum for every model.

Before we carry out the comparison between model results and sea ice cover reconstructions for the LIG period, we compare the results of the models for PI to the observations at the reconstruction sites (Fig. 4 for the comparison of annual mean sea ice concentrations and Fig. 5a and b for winter and summer). Models generally overestimate sea ice cover at the three northernmost sites in summer and in annual mean and over the seven northernmost sites for the winter season. Those sites are actually very close to the sea ice edge and the overestimation could correspond to the fact that the observations are for 1981 to 2002 period, which was already warmer than the pre-industrial one.

## 3.2 LIG sea ice

The models show a minimum monthly MMM SIA for the LIG of $3.20 \pm 1.50 \times 10^6$ km$^2$, and a maximum MMM SIA of $15.95 \pm 2.61 \times 10^6$ km$^2$. In terms of SIE, the minimum MMM extent is $5.39 \pm 2.13 \times 10^6$ km$^2$, while the maximum MMM extent is equal to $18.38 \pm 3.12 \times 10^6$ km$^2$. Thus, compared to the PI results, there is a reduction of ca. 50 % in the MMM minimum (summer) monthly SIA in the LIG results,

and of nearly 40 % in terms of SIE, but a slight increase in the winter monthly MMM SIA and SIE. Every model shows an often substantial reduction in summer sea ice between the PI and LIG.

There is a large amount of inter-model variability for the LIG SIA and SIE during the summer (Fig. 6 and Table 4). Out of the 16 models, 1 model, HadGEM3, shows a LIG Arctic Ocean free of sea ice in summer, i.e. with an SIE lower than $1 \times 10^6$ km$^2$. CESM2 and NESM3 show low SIA values (slightly above $2 \times 10^6$ km$^2$) in summer for the LIG simulation, but their minimum SIE values are below $3 \times 10^6$ km$^2$. Both HadGEM3 and CESM2 realistically capture the PI Arctic sea ice seasonal cycle. On the other hand, NESM3 overestimates winter ice and the amplitude of the seasonal cycle in SIA and SIE, while simulating realistic PI values for both SIA and SIE (Cao et al., 2018). This seasonal cycle is amplified in the LIG simulation, with an increase in SIA and SIE in winter and a decrease in summer, following the insolation forcing. Hence, the difference in the response of these models to LIG forcing in terms of sea ice does not appear to only depend on differences in PI sea ice representation.

For the winter, only one model (EC-Earth) simulates a decrease in SIA and SIE of around $2 \times 10^6$ km$^2$, two other models (ACCESS and INM-CM4-8) simulate a slight decrease in SIA and SIE, all other models simulate an increase in both SIA and SIE. All models therefore show a larger sea ice area amplitude for LIG than for PI, and the range of model results

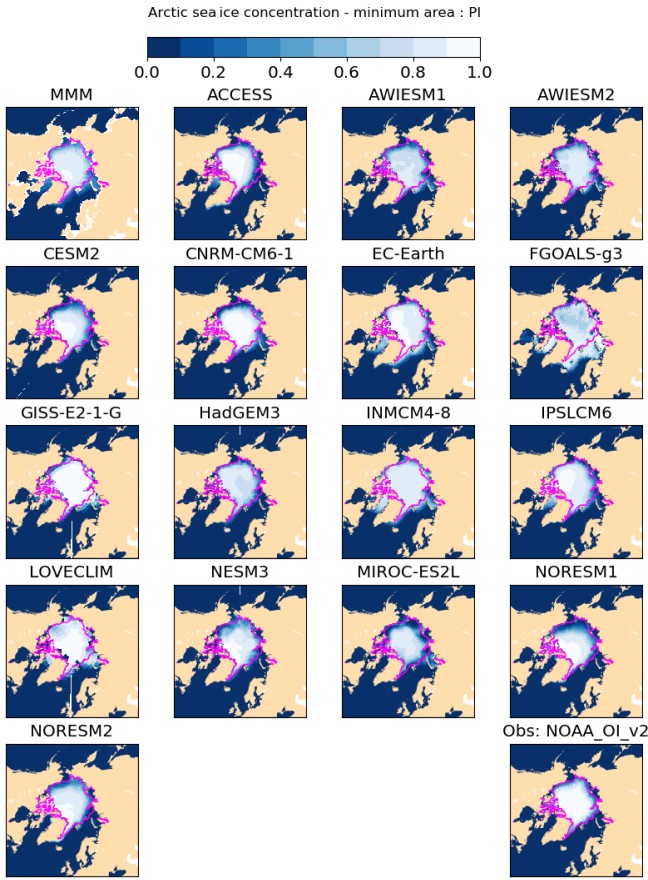

**Figure 3.** PI sea ice concentration for the month of minimum SIA as computed for Fig. 2. The magenta contour shows the 0.15 isocontour of the NOAA_OI_v2 observations (Reynolds et al., 2002, see the Data availability section) averaged over the years 1982–2001.

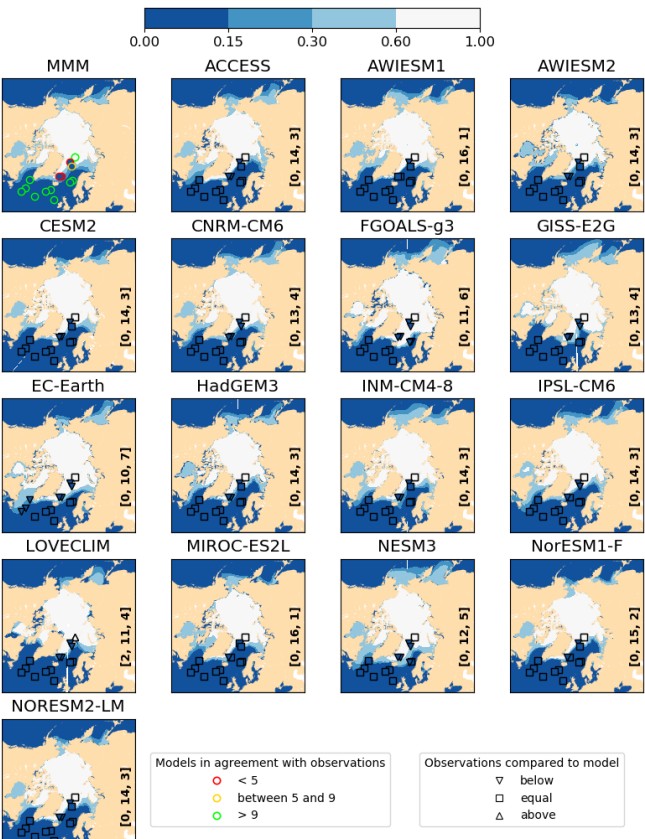

**Figure 4.** Sea ice annual concentration simulated for PI, for the multi-model mean (MMM) and for each model. The colour filling of the symbols on the maps correspond to the observed values at each site, which are classified into three categories according to the NOAA_OI_v2 dataset: perennial cover (9 to 12 months), seasonal cover (3 to 9 months) and ice-free state (0 to 3 months). On the MMM panel, for each data site the colour of the symbol outline corresponds to the number of models simulating the observed ice cover. On the panels for individual models, the shape of the symbol depends on the observed result being below the simulated one (triangle down), above the simulated one (triangle up) or in the same category as the simulated one (circle). The number of sites for which reconstructions are equal to and above the number of months simulated by models are written at the bottom-right corner of each panel.

### 3.3 LIG model–data comparison

We limit our comparison to the sites for which the chronology is good. These cores mostly show ice-free conditions in summer, except for the northernmost site (core PS92/039-2), which is at least 75 % covered by ice in summer (Fig. 5c). Two other sites at high latitude (PS213861 and PS93/006-1, for which sea ice has been reconstructed based on dinocysts and IP25/PIP25), show summer conditions which are "probably ice-free". Only four models simulate more than 75 % sea ice concentration over the northernmost site, but they also simulate more than 75 % sea ice concentration over the two following sites (in descending order of latitudes), and FGOALS-g3 simulates more than 75 % sea ice concentration for another four sites for which the reconstructions show no sea ice. On the other hand, 10 models simulate no sea ice

concentration at all over the reconstruction sites in summer, and therefore probably overestimate the LIG summer sea ice reduction. From these reconstructions, we cannot distinguish the performance of the models simulating a strong reduction of sea ice from the model simulating a nearly total disappearance of summer sea ice in the Arctic. Apart from FGOALS-g3, which simulates extensive sea ice cover for both periods, there does not appear to be a strong relationship between the PI and LIG model results over the data sites: models which simulate sea ice cover over the three northernmost sites at the LIG do not necessarily simulate large sea ice concentra-

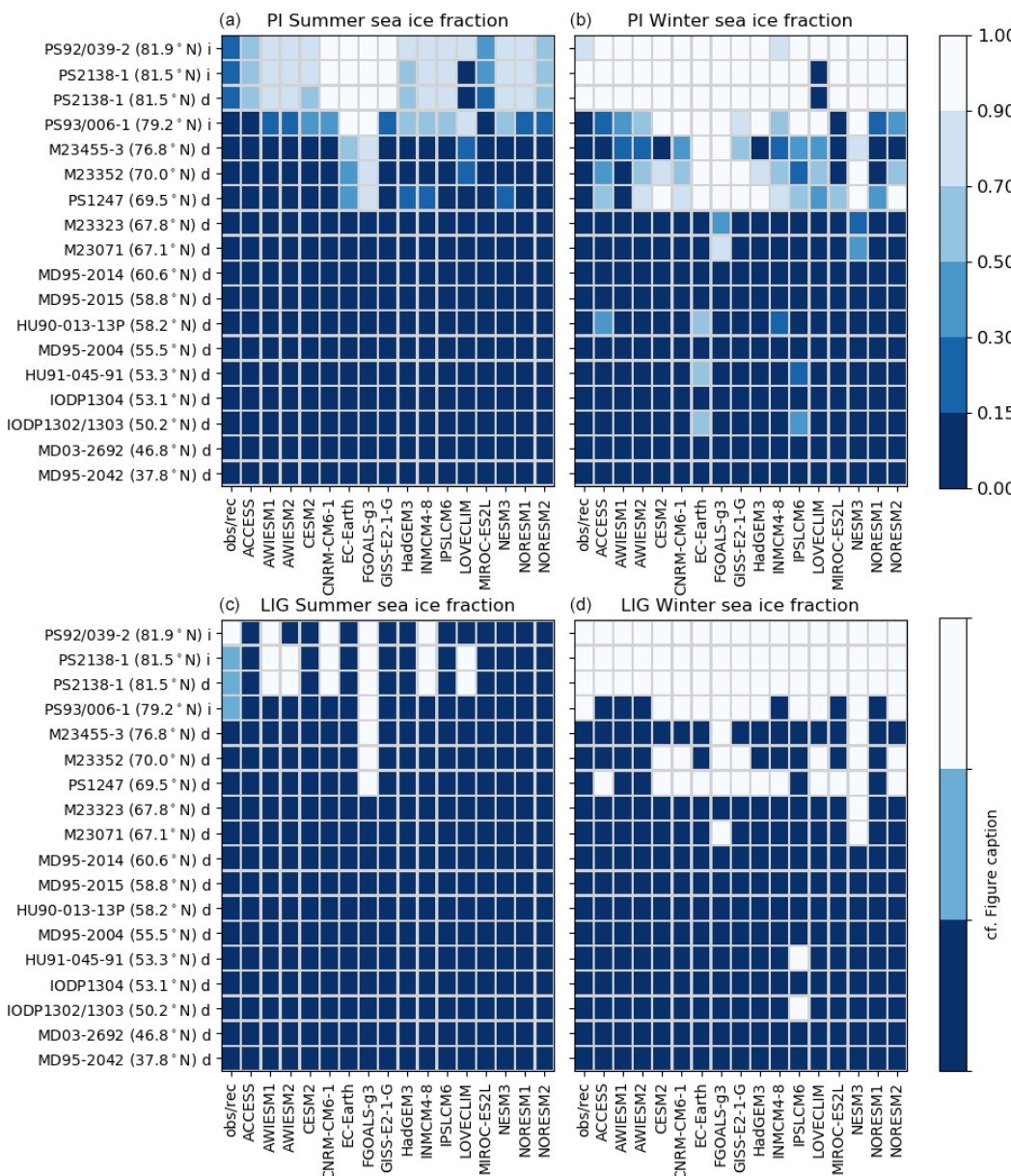

**Figure 5.** Model–data comparison as a function of latitude and record site for PI **(a, b)** and LIG **(c, d)**. For each LIG data site, the NOAA_OI_v2 observations (PI) or reconstructions (LIG) are shown in the first column of each plot, and the model results are in the columns to the right. Both the model results and the NOAA_OI_v2 observations are shown in terms of sea ice fraction averaged over the month of minimum **(a, c)** or maximum **(b, d)** Northern Hemisphere sea ice area and the previous and following months. For the LIG, the qualitative assessments (eighth column of Table 1) have been used for records with good chronological control. The letter next to the name of the site stands for indicator used for the reconstruction: dinocysts ("d") or IP25/PIP25 ("i"). For summer conditions, dark blue shading is used for the "no sea ice" category, light blue shading is used for "summer probably ice-free" conditions, and white shading is used for "summer sea ice concentration > 75 %" and "perennial sea ice". For winter, dark blue shading is used for "ice-free all year round" conditions and white shading is used for "seasonal sea ice conditions" and "perennial sea ice". The model results are averaged as they are for PI and shown following the colour scale on the right-hand side of the plots.

tions over these sites for PI (e.g. LOVECLIM, AWIESM1 and AWIESM2).

For the winter season, the reconstructions show the four northernmost sites to be ice covered. The reconstructions for most other sites are qualitatively given as "nearly ice-free

all year round" or "ice-free all year round". Model results are generally in agreement with the reconstructions for the three to four northernmost sites (Fig. 5d). Most models simulate sea ice over some of the sites characterised by "nearly ice-free all year round" conditions, and only one model (IP-

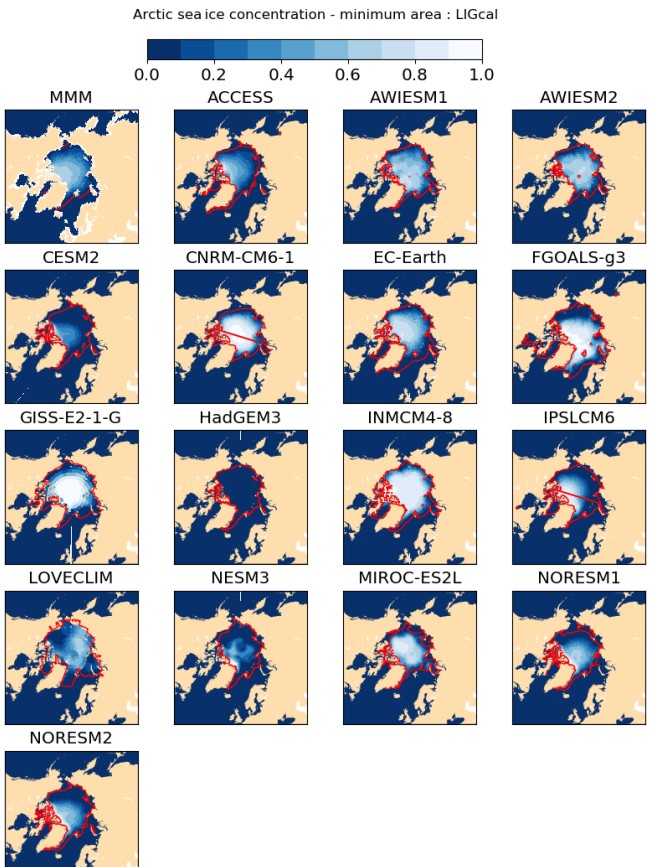

**Figure 6.** LIG sea ice concentration for the month of minimum SIA (computed with calendar adjustment) as computed for Fig. 2. The magenta contour shows the 0.15 isocontour of the corresponding PI simulation.

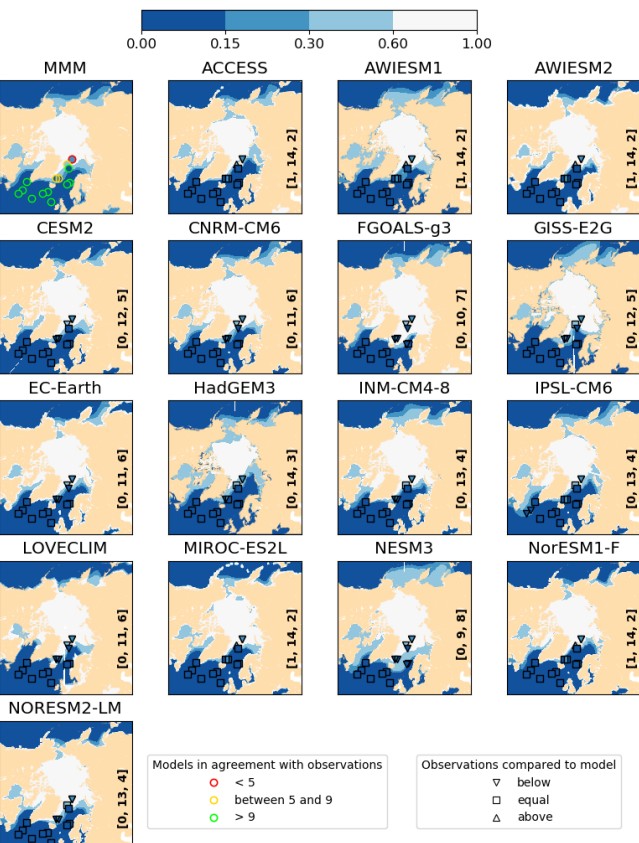

**Figure 7.** Sea ice annual concentration during the LIG (computed with calendar adjustment), for the multi-model mean (MMM) and for each model. The colour-filling of the symbols on the maps correspond to the reconstructed values, classified into three categories: perennial cover (9 to 12 months), seasonal cover (3 to 9 months) and ice-free state (0 to 3 months). On the MMM panel, for each data site, the colour of the symbol outline corresponds to the number of models simulating the reconstructed ice cover. On the panels for individual models, the shape of the symbol depends on the model result being below the reconstructed one (triangle down), above the reconstructed one (triangle up) or in the same category as the reconstructed one (circle). The number of data points which are above, equal and below the number of months simulated by models are written in the bottom-right corner of each panel.

SLCM6) simulates sea ice cover over a site for which the reconstructions show ice-free conditions. The model–data agreement is therefore quite good for the winter season. In this case, the model results for LIG appear to be related to their results for PI, with models simulating more sea ice for LIG being those simulating more sea ice for PI.

Figure 7 shows a quantitative model–data comparison in terms of annual mean sea ice concentration, which is the variable for which we have the highest number of reconstructions (Table 1). From this, we see that it is more difficult for the models to realistically capture sea ice change over the core sites near Greenland close to the sea ice edge. If we cross-compare the observation–model match for each model for both the PI (Fig. 4) and the LIG (Fig. 7) then FGOALS-g3 and NESM3 have difficulties in accurately capturing sea ice cover at the core site locations in the Nordic Seas, whilst AW-IESM1 and NORESM1-F best display sea ice cover close to the sea ice edge near Greenland and in the Nordic seas for both time periods. It is these Nordic Seas sea ice edge differences (over the core sites listed in Table 1) that make

the difference between the simulation–data matches for each model.

## 4 Discussion of model differences

Whilst we cannot yet definitely establish the most likely Arctic sea ice conditions during the LIG, we can investigate sea ice differences across models when we have sufficient model data. We have first performed this analysis for the three models for which we had sufficient data: CESM2, HadGEM3, and IPSLM6. These models each represent a distinct sea ice response to the LIG forcing, i.e. summer sea ice concentra-

tion less than 0.15 everywhere (HadGEM3), significant summer sea ice retreat with concentration less than 0.8 in the Central Arctic (CESM2) and modest summer sea ice retreat with a small area with sea ice concentration close to 1 in the Central Arctic (IPSLCM6).

Sea ice formation and melting can be affected by a large number of factors inherent to the atmosphere and the ocean dynamics, alongside the representation of sea ice itself within the model (i.e. the type of sea ice scheme used). In coupled models it can be extremely difficult to identify the causes of essentially coupled model behaviour. Nevertheless, we discuss the short-wave (SW) surface energy balance, ocean, and atmosphere circulations and comment on cloudiness and albedo changes.

### 4.1 Atmospheric energy budget differences

The atmospheric energy budget LIG–PI anomaly (Fig. 8) is negative in winter and strongly positive in summer, following the imposed insolation anomaly. These anomalies in total heat budget are dominated by the SW budget contribution from May to August. We split the SW budget into the downward (SWdn) and upward (SWup) contributions. Both fluxes are defined to be positive when they are downward and negative when they are upward. Hence, the total SW budget (in black) is the sum of the SWdn contribution (in red) and the SWup contribution (in blue). In this figure, a positive SWup anomaly means that the SWup is less intense at LIG than at PI, hence contributing to an increase in the net SW flux.

For all the models, the total heat budget anomaly is due to (i) an increased downward short-wave flux in spring resulting from the insolation forcing and (ii) a decreased upward short-wave flux in summer, related to the decrease of the albedo due to the smaller sea ice cover. During summer, this decrease in upward short-wave flux more than compensates the decrease in SWdn, which is maximum in August.

The summer anomaly reaches $80\,\mathrm{W\,m^{-2}}$ in June for HadGEM3, $60\,\mathrm{W\,m^{-2}}$ for IPSLCM6 and $50\,\mathrm{W\,m^{-2}}$ for CESM2. The differences between the model results are due to a different phasing of the SWdn and SWup anomalies for HadGEM3, compared to the other two models: for HadGEM3, the two fluxes peak in June, while for CESM2 and IPSLCM6, the SWdn flux peaks in May and the SWup signal peaks in July, and thus the anomaly in these fluxes partly compensate for each other. HadGEM3 shows a larger net SW increase despite a SWdn anomaly which is smaller than for the other two models. On the other hand, HadGEM3's SWup component is stronger and always positive, which is different to the other two models, which show a negative SWup contribution in April–May. These differences are associated with differences in albedo for the three models (Fig. 9). HadGEM3's sea ice and Arctic Ocean albedos are always smaller than those simulated by IPSLCM6 and CESM2 and the difference is larger for LIG than for PI. The albedo simulated by HadGEM3 in May and June is particu-

larly low compared to the two other models, which explains why the SWup component peaks earlier. The albedo LIG–PI anomalies over the whole Arctic show that the sea ice albedo feedback is most effective in HadGEM3.

In terms of cloudiness, IPSLCM6 shows differences in the properties of clouds, in terms of optical depth, between PI and LIG, but this could not be investigated due to a lack of data (thus far) for the other models. Thus we cannot tell if LIG–PI anomalies in SWdn fluxes, i.e. differences between HadGEM3's and CESM2 flux, also have a contribution due to cloud changes.

The comparison to other model results (Fig. 10) confirm that the behaviour of HadGEM3 is unusual in terms of energy budget. It is the only model in which the anomalies in SWup and SWdn are exactly in phase and produce a much larger anomaly in total heat budget, while in other models those anomalies are not in phase and partly compensate each other.

### 4.2 Ocean and atmosphere circulation differences

Changes in Arctic sea ice related to ocean heat transport have been found for the CESM large ensemble (Auclair and Tremblay, 2018). The differences can then be amplified by the sea ice albedo feedbacks. We check this in our models by calculating long-term means of the maximum meridional stream function at $26°\,\mathrm{N}$ for the PI and LIG simulations. These are 19.5 and 18.7 for CESM2, 15.6 and 15.8 Sv for HadGEM3, and 12.9 and 10.4 for IPSLCM6. Thus, the CESM2 and HadGEM3 models exhibit an Atlantic Meridional Overturning Circulation (AMOC) that is almost unchanged between PI and LIG, while in the IPSLCM6 model the AMOC weakens. This implies that a reduced northward oceanic heat transport could prevent sea ice loss in the Central Arctic in some but not all models (see also Stein et al., 2017).

Some differences in the response of sea ice to LIG forcing therefore appear to be due either to differences in atmospheric response (HadGEM3 vs. IPSL-CM6 and CESM2), similar to mechanisms found for current sea ice decline (e.g. He et al., 2019; Olonscheck et al., 2019) or to changes in ocean heat transport (CESM2 vs. IPSLCM6). But while AMOC changes partially explain the differences found between IPSL (more sea ice in Central Arctic) and CESM2 and HadGEM3 (less sea ice in Central Arctic), they do not explain differences between ice-free and ice-covered conditions in HadGEM3 and CESM2.

Differences in atmospheric circulation changes could also explain difference in sea ice response to LIG forcings. We therefore investigate LIG–PI anomalies in sea level pressure (Fig. 11). Most models simulate a decrease in summer mean sea level pressure largely encompassing the Arctic Ocean and adjacent continents. This decrease is not as strong over the Nordic Seas as it is over the Arctic, and this local heterogeneity over the Nordic Seas is model dependent. However, the anomaly in atmospheric circulation is more zonal over the

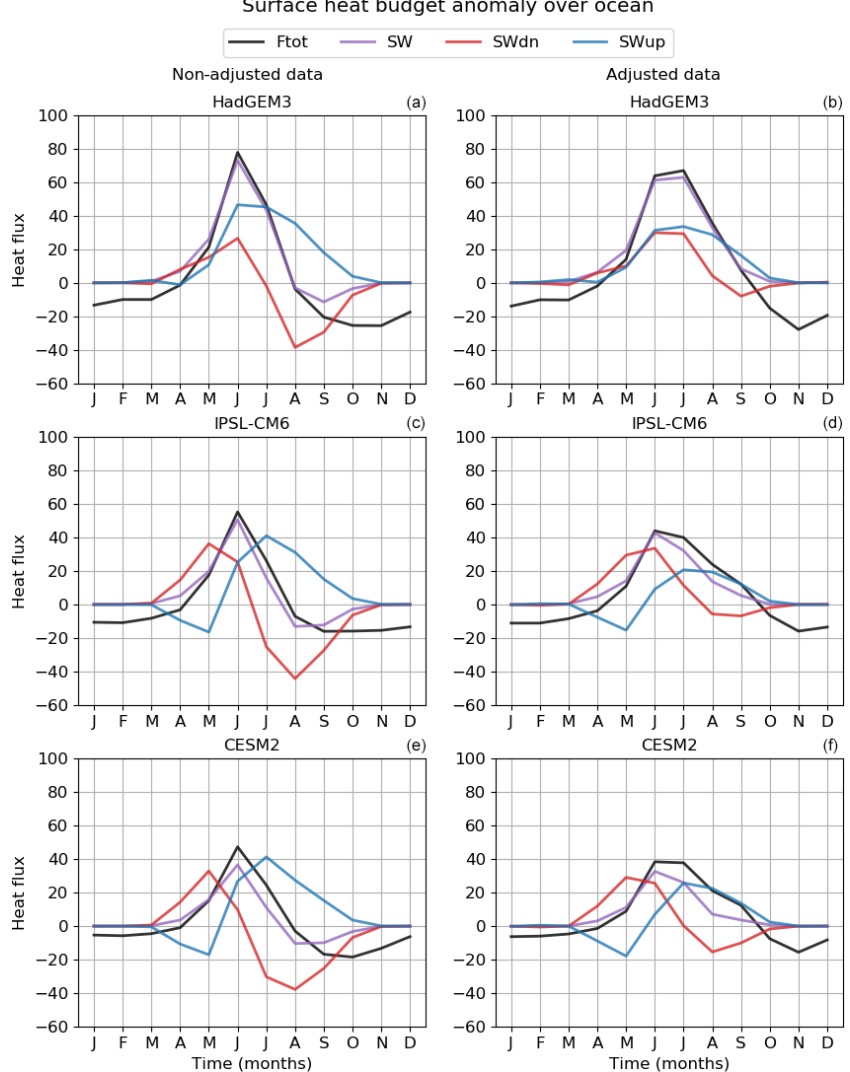

**Figure 8.** The main components of the atmospheric energy budget at the surface averaged over the Arctic (70–90° N) for HadGEM3, CESM2 and IPSLCM6. The LIG–PI anomalies as shown as a function of the month for the total energy budget (Ftot, black), the SW budget (SW, violet), and for the downward (SWdn, red) and upward SW (SWup, blue) fluxes. The sign convention for all fluxes is that fluxes pointing downward are positive and fluxes pointing upward are negative. Panels **(b)**, **(d)** and **(f)** show results for which the LIG calendar has been taken into account (for the LIG simulations), while panels **(a)**, **(c)** and **(e)** show the results averaged on the PI calendar both for PI and LIG.

Nordic Seas and northern North Atlantic in HadGEM3 than in CESM2 or IPSLCM6, and therefore differences in atmospheric circulation are probably not causing more warm air to enter the Arctic for HadGEM3 and are thus not the cause of HadGEM3 being so warm over the Arctic. The mean sea level pressure winter anomaly is characterised by a deepening of the Icelandic low for all models except NESM3.

Other factors that remain to be investigated include clouds and ocean heat uptake in the Arctic in the different models, e.g. as a function of stratification.

## 4.3 Transient CO$_2$ forced responses: LIG vs. transient 1pctCO2

The LIG has higher insolation than PI at high northern latitudes during spring and summer and less significant changes in winter insolation. This is distinct from the increased GHG, which is the dominant forcing for future climates. However, since sea ice minimum occurs in summer, it is of interest to consider possible relationships between CMIP6 model responses for the LIG and those for the transient 1pctCO2 experiments. A total of 12 models have the LIG, PI and 1pctCO2 simulations available. These include models with large, small and intermediate responses in sea ice for the LIG.

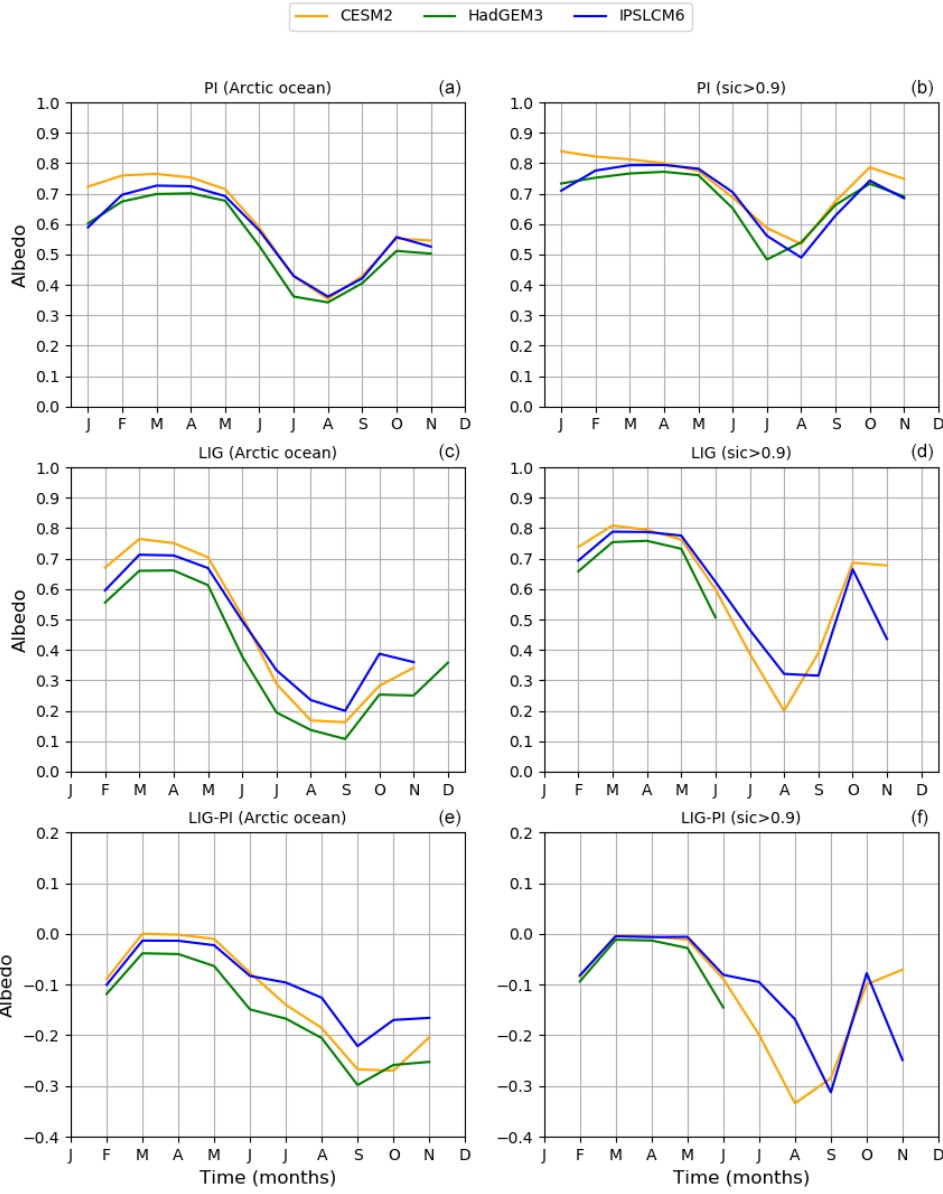

**Figure 9.** Albedo over the Arctic for PI **(a, b)**, LIG **(c, d)** and LIG–PI **(e, f)** for HadGEM3, IPSL-CM6 and CESM2. The albedo has been recomputed from the SWup and SWdn fluxes. Panels **(a)**, **(c)** and **(e)** show the results for the whole Arctic, while panels **(b)**, **(d)** and **(f)** show the results for areas where the sea ice fraction is larger than 0.9. All LIG values have been calendar adjusted.

Figure 12 suggests that there is indeed such a relationship between the summer sea ice concentration decreases for LIG and the averages from years 50 to 70 of the transient 1pctCO2 simulations: the models that respond strongly at the LIG also respond strongly for the 1pctCO2 forcing, and the model with the smallest response for the LIG (INMCM4-8) has the smallest response to the 1pctCO2 forcing. The relationship shown in Fig. 12 does not last for later periods in the 1pctCO2 runs, when the winter sea ice is also affected by the increased greenhouse gas forcing. This implies intercomparisons between the LIG simulation and simulations

with moderate $CO_2$ increase (during the transition to high $CO_2$ levels) should be investigated.

## 5   Conclusions

The Last Interglacial period (LIG) was the last time global temperature was substantially higher than the pre-industrial period at high northern latitudes (Otto-Bliesner et al., 2013; Capron et al., 2017; Otto-Bliesner et al., 2017; Fischer et al., 2018; Otto-Bliesner et al., 2020). To help understand the role of Arctic sea ice in these changes, we present a new synthesis

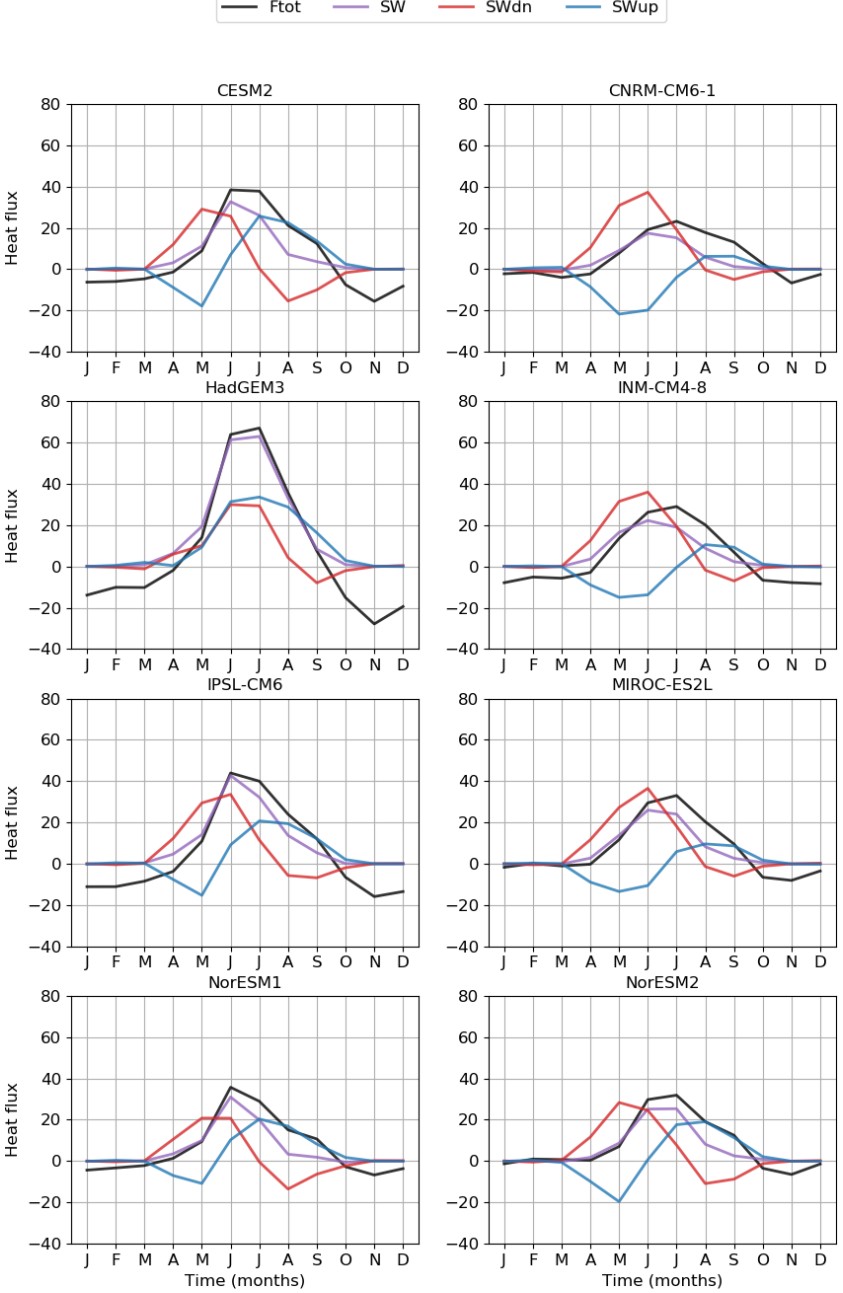

**Figure 10.** The main components of the atmospheric energy budget averaged over the Arctic (70–90° N) for HadGEM3, CESM2, CNRM-CM6-1, IPSLCM6, INM-CM4-8, MIROC-ES2L, NorESM1 and NorESM2. The LIG–PI anomalies are shown as a function of the month for the total energy budget (black), the SW budget (violet), and for the downward (red) and upward SW (blue) fluxes. The sign convention for all fluxes is the same as for Fig. 8.

of LIG sea ice information using marine core data collected in the Arctic Ocean, Nordic Seas and northern North Atlantic and compare this to PMIP4-LIG simulations.

Our synthesis shows that south of 79° N in the Atlantic and Nordic seas the LIG was definitely seasonally ice-free. These southern sea ice records provide quantitative esti-mates of sea surface parameters based on dinoflagellate cysts (dinocysts). North of 79° N the sea-ice-related records are more difficult to obtain and interpret. However, the core at 81.5° N brings evidence of summer being probably season-ally ice-free during the LIG from two indicators: dinocysts and IP25/PIP25. The northernmost core with good chronol-

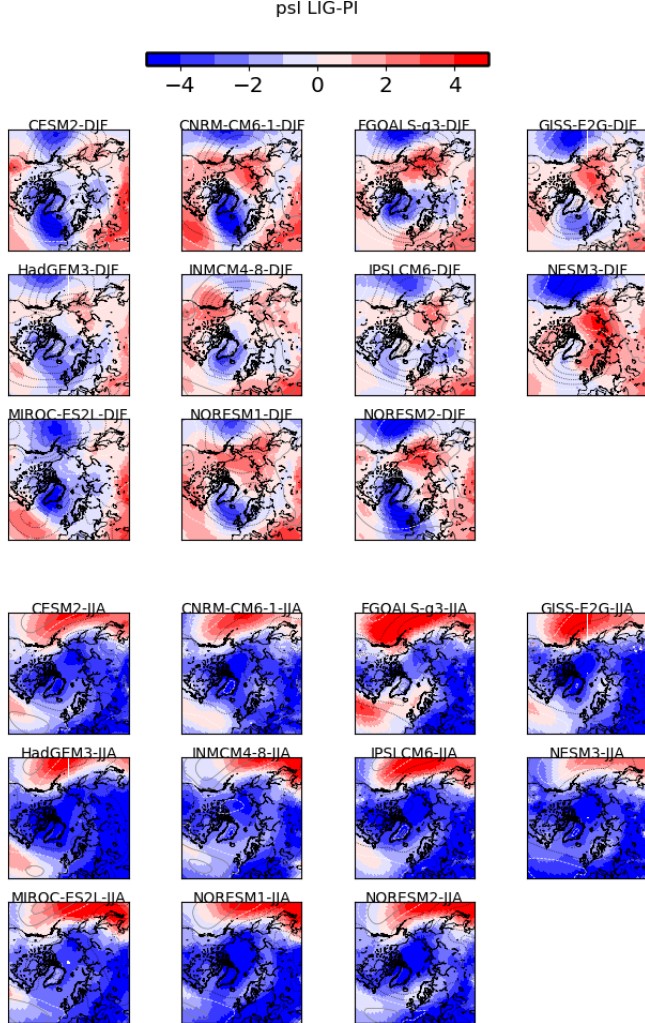

**Figure 11.** Anomalies (LIG–PI, hPa, shading) in mean sea level pressure for DJF (top plots) and JJA (bottom plots). Contours indicating PI values are superimposed: values every 5 hPa, 1005 hPa isobar in white, black contours for lower values and grey contours for higher values.

ogy is located at 81.9° N and shows evidence of substantial (> 75 %) sea ice concentration all year round. Other cores, with debated chronologies, have not been used for model–data comparisons in the present study.

Model results from 16 models show a multi model mean (MMM) summer SIA LIG of $3.20 \pm 1.29 \times 10^6$ km², and a winter monthly MMM area of $15.95 \pm 1.21 \times 10^6$ km². This is a reduction in SIA of 50 % for the minimum summer month between the PI and LIG but almost no change for the winter month MMM. Every model shows an often substantial reduction in summer sea ice between the PI and LIG. For the winter, only one of the 16 models shows a (small) winter reduction in sea ice between the PI and LIG. This reinforces that the key seasons for understanding LIG warming are the spring, summer and autumn.

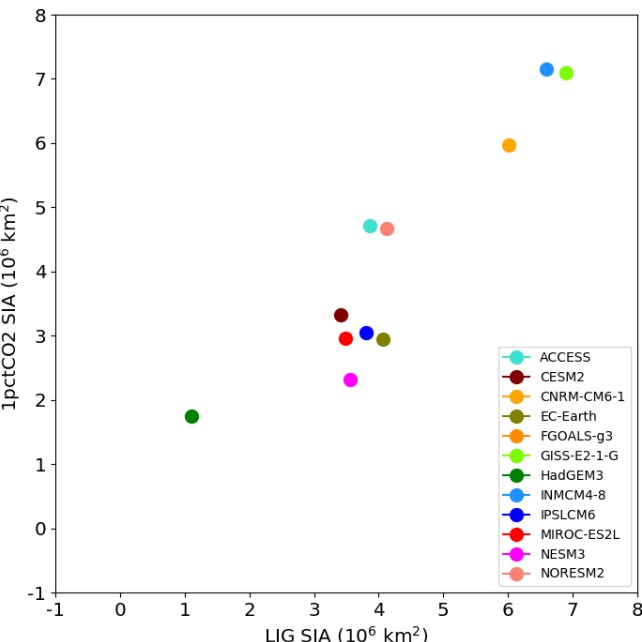

**Figure 12.** LIG vs. 1pctCO2 July–August–September sea ice areas (for sea ice concentrations larger than 0.15). The results for the 1pctCO2 simulations have been averaged for years 50 to 70.

We investigate reasons for inter-model differences in LIG Arctic sea ice simulations: we find that the LIG total heat budget anomaly in the Arctic is due to (i) an increased downward short-wave flux in spring, resulting from the insolation forcing, and (ii) a decreased upward short-wave flux in summer, related to the decrease of the albedo due to the smaller sea ice cover. During summer, this decrease in upward short-wave flux more than compensates the decrease in the SWdn, which is at a maximum in August. Differences between the model results are due to a different phasing of the up and down short-wave anomalies in the different models and are associated with the differences in model albedo.

Analysis of IPSLCM6 results shows differences in the properties of clouds, in terms of optical depth, between PI and LIG. Further work is required to identify if this is also important for other models. Changes in Arctic sea ice may also be related to ocean heat transport. Here, we have shown that ocean circulation changes occur for some (but not all) LIG simulations. Other factors that remain to be investigated include clouds and ocean heat uptake in the Arctic in the different models.

Most models agree with the reconstructed year-round ice-free northern North Atlantic. Model–data disagreement for the LIG occur over the Nordic Seas, close to Greenland and at the boundary with the Arctic Ocean, where many models overestimate annual mean sea ice concentration. This is not fully related to the model performance for summer. Indeed, 12 of 16 models simulate little sea ice cover over the north-ernmost site and 10 of the models simulate less than 25 %

sea ice concentration over the site at 81.5° N. It is not possible, from the available data, to decide on the best models, in particular in terms of summer sea ice. The northernmost site appears to discriminate those models that simulate very little sea ice at this site. However, models which do simulate > 75 % summer sea ice concentration at this site also simulate > 75 % summer sea ice concentration for the two sites at 81.5 and 79.2° N, just south of the northernmost site, which is not realistic. More reconstructions with good chronology are needed in the Central Arctic to determine which model behaviour is more realistic, and in particular if the summer ice-free Arctic simulated by the HadGEM3 model alone, among the 16 models, is possible. This would be key in assessing ESMs used for future projections with respect to climates with much warmer summers than today. This means that it is all the more crucial that there appear to be a nearly linear relationship between the ESM simulations of summer sea ice for the near future (years 50 to 70 of transient 1pctCO2 simulations) and that simulated for the LIG: the models which respond strongly to the LIG forcing also respond strongly for the 1pctCO2 forcing. This implies inter-comparisons between the LIG simulation and simulations with a moderate $CO_2$ increase (during the transition to high $CO_2$ levels) may yield insight into likely 21st century Arctic sea ice changes, especially if we achieve a more extensive characterisation of LIG Arctic sea ice from marine cores.

**Data availability.** The NOAA Optimum Interpolation (OI) V2 dataset for sea ice concentration has been retrieved from https://psl.noaa.gov/data/gridded/data.noaa.oisst.v2.html TS8 . The dataset used for the present study is the monthly dataset: ftp://ftp.cdc.noaa.gov/Datasets/noaa.oisst.v2/icec.mnmean.nc TS9 .

The original output data from the model simulations used in this study are available from the Earth System Grid Federation (https://esgf-node.llnl.gov/ TS10 ), the data repository for CMIP6 simulations, or on open repositories, as listed in this paper's companion paper by Otto-Bliesner et al. (2020).

Nonetheless, the exact data shown in Figs. 2 to 12 are also provided as a Supplement to this paper: the numbers are given as text files for Figs. 2, 8, 9, 10 and 12. NetCDF files are provided with the data shown on Figs. 3, 4, 6, 7 and 11. For each model, there is one netCDF file (*modelname*_sea-ice-diags_cp-2019-165.nc) with sea ice variables (Figs. 3, 4, 6 and 7) and one netCDF file (*modelname*_psl_cp-2019-165.nc) with the mean sea level pressure (Fig. 11).

**Supplement.** The supplement related to this article is available online at: https://doi.org/10.5194/cp-16-1-2020-supplement.

**Author contributions.** MK, LCS, MS, MVG, RS and AdV are joint first authors for this paper. MK and LCS planned the study with the other QUIGS members. MK and MS analysed all model simulations and produced all model figures. LCS wrote the paper. MVG contributed substantially to the first draft and compiled all model information. AdV, IMV, RS and LCS compiled the sea ice dataset, and IMV produced the dataset map. DS co-planned some of the model analysis. MK and all other here-unnamed authors contributed model data. DS, DF, CB, and JS provide sea ice modelling advice. All authors read the draft and commented on the text. TS11

**Competing interests.** The authors declare that they have no conflict of interest.

**Special issue statement.** This article is part of the special issue "Paleoclimate Modelling Intercomparison Project phase 4 (PMIP4) (CP/GMD inter-journal SI)". It is not associated with a conference.

**Acknowledgements.** We acknowledge the QUIGS (Quaternary Interglacials working group endorsed by PAGES and PMIP) for making this comparison possible, thanks in particular to the workshop organised by this group in Cambridge, UK, in July 2019. We are grateful to the World Climate Research Programme, which, through its Working Group on Coupled Modelling, coordinated and promoted CMIP6. We thank the climate modelling groups for producing and making their model output available, the Earth System Grid Federation (ESGF) for archiving the data and providing access to them, and the multiple funding agencies who support CMIP6 and ESGF. The Paleoclimate Modelling Intercomparison Project is thanked for coordinating the lig127k protocol and making the model–model and model–data comparisons possible within CMIP6. PMIP is endorsed by WCRP and CLIVAR. We also acknowledge NOAA/OAR/ESRL PSD, Boulder, Colorado, USA, for their optimally interpolated sea ice product, downloaded from their website at https://www.esrl.noaa.gov/psd/ TS12 . Qiong Zhang acknowledges the HPC resources provided by the Swedish National Infrastructure for Computing (SNIC) at the National Supercomputer Centre (NSC). Bette L. Otto-Bliesner, Esther C. Brady and Robert A. Tomas acknowledge the CESM project, which is supported primarily by the National Science Foundation (NSF). This material is based upon work supported by the National Center for Atmospheric Research (NCAR). Computing and data storage resources, including the Cheyenne supercomputer (https://doi.org/10.5065/D6RX99HX, Computational and Information Systems Laboratory, 2019), were provided by the Computational and Information Systems Laboratory (CISL) at NCAR. The Last Interglacial studies of Anne de Vernal have been supported by the TS13 Natural Sciences and Engineering Research Council of Canada and the "Fonds de recherche du Québec – Nature et technologies". The NorESM simulations benefitted from resources provided by UNINETT Sigma2 – the National Infrastructure for High Performance Computing and Data Storage in Norway. The ACCESS-ESM 1.5 experiments were performed on Raijin at the NCI National Facility at the Australian National University, through awards under the National Computational Merit Allocation Scheme, the Intersect allocation scheme, and the UNSW HPC at NCI Scheme. LOVECLIM experiments were performed on UNSW HPC Katana. Ayako Abe-Ouchi, Ryouta O'ishi, and Sam Sherriff-Tadano thank JAMSTEC for use of the Earth Simulator supercomputer.

**Financial support.** This research has been supported by the PAlaeo-Constraints on Monsoon Evolution and Dynamics (PACMEDY) Belmont Forum project (grand no. 01LP1607A), German Federal Ministry of Education and Science (BMBF) PalMod II WP 3.3 (grant no. 01LP1924B), NERC (projects NE/P013279/1 and NE/P009271/1), European Union's Horizon 2020 research and innovation programme (grant agreement no. 820970), "Convention des Services Climatiques" from IPSL, Russian state (assignment project 0148-2019-0009), RSF (grant no. 20-17-00190), NSF (cooperative agreement no. 1852977), Australian Research Council (grant nos. FT180100606 and DP180100048), Arctic Challenge for Sustainability (ArCS) Project (grant no. JPMXD1300000000), Arctic Challenge for Sustainability II (ArCS II) Project (grant no. JPMXD1420318865), JSPS KAKENHI (grant no. 17H06104), MEXT KAKENHI (grant no. 17H06323), Natural Sciences and Engineering Research Council of Canada, and the "Fonds de recherche du Québec – Nature et technologies".

**Review statement.** This paper was edited by Marie-France Loutre and reviewed by Julie Brigham-Grette and one anonymous referee.

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

## Remarks from the language copy-editor

CE1   Please note that if this is two affiliations "Geotop and the Departement...", this will have to split.

## Remarks from the typesetter

TS1   Since affiliation 19 was just replaced (instead of a new affiliation added), the numbering has been left as is here.

TS2   Please check: should 11 really be updated to 12?

TS3   This reference is not in the reference list. Please add it.

TS4   Please provide definition for N/A.

TS5   According to our standards, changes like this must first be approved by the editor, as data have already been reviewed, discussed and approved. Please provide a detailed explanation for the requested changes in this table that can be forwarded to the editor. Please note that this entire process will be available online after publication. Upon approval, we will make the appropriate changes. Thank you for your understanding.

TS6   According to our standards, changes like this must first be approved by the editor, as data have already been reviewed, discussed and approved. Please provide a detailed explanation for the requested changes in this table that can be forwarded to the editor. The best way would be to prepare an updated Table 4 to make it easier for the editor to check and confirm the new values.

TS7   Please provide a definition for na.

TS8   Please provide a reference list entry including creators, title, and date of last access.

TS9   Please provide a reference list entry including creators, title, and date of last access.

TS10  Please provide a reference list entry including creators, title, and date of last access.

TS11  It is our house standard to list the initials of all contributing authors at least once. Please expand any contributions (regardless of length) to meet this standard. Thank you.

TS12  Please provide date of last access and consider adding this information to the code and data availability section.

TS13  This information has been added to the financial support section, too. Please check if it can be removed from the acknowledgements section.

TS14  Please provide all author names and make sure that all authors are listed in the correct order: last name, initial(s).

TS15  Please provide page range or article number with DOI.

TS16  This reference is not cited in the text. Please check.

TS17  As long as the final version is not published, this paper needs to be cited as "in review".