# Peer review of "A multi-model CMIP6 study of Arctic sea ice at 127 ka: Sea ice data compilation and model differences"

_Climate of the Past, 2019_

## Author Comment (AC1) · 23 Jan 2020

Table 2 contained a mistake for core PS2138-2. The numbers should read as follows: sea ice cover months mininum:1 maximum: 6 sea ice annual concentration minimum: 0.1 maximum: 0.3. We apologize for this mistake. The figures using the data from table 2 use the correct values.

[Figure]

---

## Referee Comment (RC1) · Julie Brigham-Grette (Referee) · 20 Feb 2020

Review of CP-2019-165 version 2 15 February 2020

As part of the Paleoclimate Modeling Intercomparison project, the purpose of this paper within the 6th phase of the coupled model intercomparison project (CMIP6) is to review the results from 12 climate models in terms of Arctic sea ice. The point of the project is to compare how the models produce Arctic sea ice during the Last Interglacial (LIG). I would like to say up front that what I most enjoyed about this paper is the honesty expressed in the evaluation. They admit that they cannot accurately state what sea ice was like during the last interglacial, but they can frankly say how the models compare.

I make suggestions to make the paper a bit more accessible to non-modelers like me. I have added comments to the pdf and created a comment summary.

The results show a wide range of minimums for summer sea ice but the mean of the 12 models suggests a 59% reduction in summer sea ice; they found that winter sea ice extend was about the same in LIG compared to Pre-Industrial (PI) (not thickness mind you, only extent). For "ground truth" they used only sediment cores from the Arctic Ocean and Fram Strait region (Table 1) with sea ice presence or absence heavily weighted toward proxies like dinocycsts and IP25, both of which have large errors associated with them. For example, the calibration of the dinocycsts for sea ice used 1955 to 2012 (page 6) and the error of prediction is +/- 12%. So one has to propagate the error in the proxies along with the differences in the models to compare with the same data.

One strong point for the results of this paper is that all version 6 models focused on a uniform set of model experimental protocols, because version 5 failed to do this and the results were more difficult to evaluate. For this simulation, they used sea ice base line for 1982 to 2001 (this is what they consider most realistic for PI), given that most remote sensing of sea ice started in 1979. True preindustrial sea ice extent and thickness can only be judged from historical data.

More detailed comments:

Comment 1, page 4: The decrease in summer sea ice is also supported by the migration of treeline documented by Lozhkin and Anderson 1995 showing range extensions of 600km for many tree species; treeline was north of the Brooks Range and similar extensions are shown in some sites on Baffin Island containing 80% birch pollen. One could go on but the paper is focused on ocean records. Note that the models used in this paper do include land surface processes, but only 2-3 models allow interactive vegetation (shown in Table 2). Comment 2, Page 10: It is now pretty widely accepted that Greenland gave up at least 2 meters of sea level equivalent during MIS 5e (LIG).
Dorthe Dahl-Jensen supports this now which is significant! So it would seem to make much more sense that CMIP6 should use the best current configuration of a 5e Greenland Ice sheet. I suggest for clarity that the authors here include an explanation why CMIP6 is not using smaller ice sheets. Lots of examples like The Cryosphere Discuss, https://doi.org/10.5194/tc-2018-225; Stone et al, 2013 C-P; Helsen et al. 2013 also published in C-P. Comment Figure 2 – Add axes labels to all boxes. Increase the font on the key, there is plenty of space for that.

Comment 3 page 14 – I could be wrong but what about propagating the error from the proxies given that dinos are +/_ 12%. Evaluate the proxy error vs the model comparison miss match?

Page 15 – You start here using 1pctCo2 for the first time. Please add something to explain this, like... Idealized 1% per year increase in Atmospheric CO2? Etc etc. Remember that not everyone reading this is a modeler so this term should be defined and add why its important.

Page 16 – consider this additional important point. Low sedimentation rates in the Arctic Ocean also means that the proxy resolution from the cores you are using are low enough to be missing 1000 yr intervals of no summer sea ice etc. These limitations may also complicate or explain the mixed messages from the 12 models. You should add this to the discussion – a few sentences.

Smaller comments are attached to the text using comment boxes. See the file named "Supplement" for details and picky editorial comments. I suggest this paper be published with minor revisions.

Please also note the supplement to this comment:
https://www.clim-past-discuss.net/cp-2019-165/cp-2019-165-RC1-supplement.pdf

---

## Referee Comment (RC2) · Anonymous Referee #2 · 3 Mar 2020

Kageyama et al. present the results of CMIP6-PMIP4 LIG simulations from 12 models and analyse them in terms of Arctic sea ice changes. They also present a new compilation of LIG sea-ice proxy data which they compare the model results with. While the discrepancies between simulations and proxy data, as well as within proxy data and within simulations, prevent any unambiguous identification of LIG sea-ice changes, the author provide valuable insights into the parameters that may influence sea-ice dynamics through their analysis of inter-model differences.

I find the manuscript well-structured and written in a concise and convincing way, and I only have minor concerns about how the proxy reconstructions were transferred into

values of sea-ice concentration and duration (as described below). I thus recommend this manuscript for publication with minor revisions.

—

Major comments about proxy data (mostly section 2.2):

I really like the author's cautious approach to provide common and clear definitions, based on sea-ice cover duration and sea-ice cover concentration, of ice-free / seasonal / perennial sea ice that facilitate data-data and model-data comparison. However, it is not always clear for me how such values have been obtained for the proxy data:

- For dinocysts: the explanation is very clear, but I miss the info on how the min and max values have been obtained (are those the min and max values of the 5 (?) best analogues? The minimum and maximum monthly sea-ice cover durations? The range of variability within the LIG time slice? Other?).

- For other proxies:

> I understand the authors attributed values of 0.15 and 0.95 for min and max sea-ice concentration at sites where sea ice was interpreted to be perennial, but I miss the info on how those values were defined for other sea-ice categories (or what are the sea-ice states corresponding to the 3 other min-max SIC combinations: 0.3-0.95, 0.3-0.6 and 0.1-0.3).

> The rationale for the attribution of min and max sea-ice cover durations is also not clear to me (in section 3.3 the authors mention they "define perennial sea ice to have at least 9 months of coverage", but I am confused because sites with min-max sea-ice concentrations of 0.15-0.95 have either min-max sea-ice cover durations of 9-12 mth/yr for IP25 or 3-11 mth/yr for faunas).

- Regarding the sites with PIP25-based interpretations, have the attributed min-max range of sea-ice concentration values been compared to some sea-ice concentration quantifications based on the calibrations recently proposed

(Xiao et al., 2015, http://dx.doi.org/10.1016/j.gca.2015.01.029; Smik et al., 2016, http://dx.doi.org/10.1016/j.orggeochem.2015.12.007) to see if both methods yield rather similar results?

- Could it be specified in Table 1 whether it is IP25 and/or PIP25?

—

Minor comments:

- At my first reading (but not the following), it was not always clear whether it was referred to sea-ice cover duration, concentration or simply sea-ice cover. Maybe SIC, SICc and SICd abbreviations (or something similar) could be used to help with this?

- L18: what is 21C?

- L69: "PI" abbreviation used as "pre-industrial" but defined as such only L133.

- Table 1: maybe it would be clearer to specify "duration" in the "sea-ice cover" column (or cf. my first minor comment), as well as "per year" for the unit.

- L90: The error of prediction for sea-ice cover concentration is indicated, but not that for sea-ice duration.

- Table 2: some info missing for CESM2 boundary conditions and LIG simulation length, LOVECLIM1.2 physical core components and LIG simulation length, and NESM3 boundary conditions and LIG simulation length.

- L134: GHG abbreviation not defined

- L156: should the reference to Figure 3 here be to Table 4 instead, as Figure 3 is referred to 4 lines after when talking about the "The detail of the geographical distribution of sea ice"?

- L175-177: maybe it would be clearer to mention that the reduction is "between the PI and LIG" in the first sentence rather than/in addition to in the second sentence (as

done in the conclusion).

- L178: 12 rather than 13 models?

- L178-181: maybe specify that the third model is NESM3 and refer to the section above regarding the reason why it does not realistically capture the PI Arctic?

- L188-191 and Figures 6 and 7: "the reconstructed values, classified into 3 categories: perennial cover (9 to 12 months), seasonal cover (3 to 9 months), ice free state (0 to 3 months)" –> how were the reconstructions based on the same proxy with e.g. (from Table 1) 3 to 12 months/year classified? Does "ambiguous interpretations" (here but I also mean in general in the MS) refer to those from the same core/area and based on different proxies (which is what I understand) or does it also refer to reconstructions from the same core and same proxy? If so, maybe it would be worth clearly mentioning it too, as it also plays a role in the difficulty to compare model and data (and highlight the proxy limitations from this other perspective) and in model-data discrepancies.

- L191: the first "to" may be removed in "it is not possible to for any one model to match"

- L192: maybe something like "comparison between the PI and LIG model *results* and PI and LIG sea ice *proxy* data" would be clearer

- L209-210: I understand that the authors do not want to solve this here, and I think it is not necessary as the focus of this paper is on the models. That said, given the proxy and model dataset presented here, and the authors being one of the world experts on these proxies, I have to admit that I was kind of expecting / hoping for this initially... :-)

- L213: "we" instead of "to"

- L224: SW abbreviation not defined

- L276: no need to redefine LIG abbr.

- L281: I would maybe rather say "These southern sea ice records are (or "correspond to" or equivalent) quantitative estimates based on dinoflagellate cysts (dinocysts)" to

avoid confusion.

- L288-289: there has been a shortcut, "periods" does not refer to anything here.

- L299: 12 models + no need to redefine MMM abbr.

- L305-306: needs to be rephrased

- Figure 1: the colour code for the cores is missing + why are there only some cores labelled? If this is a matter of space, numbers could be used in Table 1 and Figure 1.

---

## Short Comment (SC1) · 13 Mar 2020

This paper makes a very important contribution to the quest to understand Arctic Sea-Ice during the LIG, contains interesting diagnostics, a great model-data comparison, and I enjoyed reading it. Because this topic is so important, I'd like to request a few extra figures that may help us understand the models and the LIG Arctic sea ice better. Is there any chance you can produce (perhaps in the supplement?) any of the following variables for the various models for the pre-industrial and LIG? In order of preference:

- Sea-ice drifts or surface currents: Was the surface circulation at the LIG different from today and what are the models showing for the PI? Do they have a realistic PI

circulation?

- Winds or windstress over the ocean: This will drive sea-ice export as well as influence sea-ice distributions. It would be interesting to see if there are any consistent changes between the PI and LIG that could be useful in interpreting the data or model differences.

- The barotropic streamfunction or upper layer circulation if different from the surface: Are the models getting the Arctic gyres right and do these gyres systematically change at the LIG? This may also indicate what may happen in the Atlantic layer which is thought to affect sea-ice, at least in the Barents Sea.

- SLP patterns: This should be a straight forward plot indication atmospheric circulation differences between the different models and between the PI and LIG

A point I'd like to challenge is that models produce a similar response to future CO2 warming and the LIG forcing (Figure 10). If one takes away the one outlier (INMCM4-8) then the correlation breaks down. The response to 1pctCO2 is much stronger in some models than others model, while the response to LIG forcing is similar in the models.

Figure 5 is very interesting and I keep coming back to it but find it very difficult to dissect what the different models are showing because the symbols overlap to much or are hidden behind the lines.

Thanks for a great paper and I am hoping you will have he means to make some of these clarifying figures. Agatha de Boer

---

## Author Comment (AC2) · 15 May 2020

**Response to reviewers**

The reviewers' comments reproduced in black, our response is given in blue.

**Preambule**

In revising our manuscript, we considered the reviewers' comments, but we have also:

- updated the results to include 2 more models for which the PMIP4-CMIP6 data became available on ESGF;

- presented our results in terms of sea ice area as well as sea ice extent, for them to be comparable to previous CMIP5 work presented in the fifth assessment report of the IPCC, and to the work on CMIP6 carried by SIMIP (Sea ice Model Intercomparison Project). The definition of these quantities is added to the manuscript for clarity.

- included a discussion on the "calendar" effect (in Section 3.2).

- verified and updated the reconstructions, in particular by being more cautious with data from Arctic cores having uncertain  chronology, but also in terms of interpretation and updated the data-model comparisons accordingly.

Our detailed response to the reviewers follows.

**Response to Julie Brigham-Grette**

As part of the Paleoclimate Modeling Intercomparison project, the purpose of this paper within the 6th phase of the coupled model intercomparison project (CMIP6) is to review the results from 12 climate models in terms of Arctic sea ice. The point of the project is to compare how the models produce Arctic sea ice during the Last Interglacial (LIG). I would like to say up front that what I most enjoyed about this paper is the honesty expressed in the evaluation. They admit that they cannot accurately state what sea ice was like during the last interglacial, but they can frankly say how the models compare.

I make suggestions to make the paper a bit more accessible to non-modelers like me. I have added comments to the pdf and created a comment summary.

The results show a wide range of minimums for summer sea ice but the mean of the12 models suggests a 59% reduction in summer sea ice; they found that winter sea ice extend was about the same in LIG compared to Pre-Industrial (PI) (not thickness mind you, only extent). For "ground truth" they used only sediment cores from the Arctic Ocean and Fram Strait region (Table 1) with sea ice presence or absence heavily weighted toward proxies like dinocycsts and IP25, both of which have large errors associated with them. For example, the calibration of the dinocycsts for sea ice used 1955 to 2012 (page 6) and the error of prediction is +/- 12%. So one has to propagate the error in the proxies along with the differences in the models to compare with the same data.

One strong point for the results of this paper is that all version 6 models focused on a uniform set of model experimental protocols, because version 5 failed to do this and the results were more difficult to evaluate. For this simulation, they used sea ice base line for 1982 to 2001 (this is what they consider most realistic for PI), given that most remote sensing of sea ice started in 1979. True preindustrial sea ice extent and thickness can only be judged from historical data.

We wish to thank Julie Brigham-Grette for her helpful comment. To us, it is important that this manuscript can be understood by a wide audience.

More detailed comments:

Comment 1, page 4: The decrease in summer sea ice is also supported by the migration of treeline documented by Lozhkin and Anderson 1995 showing range extensions of 600km for many tree species; treeline was north of the Brooks Range and similar extensions are shown in some sites on Baffin Island containing 80% birch pollen. One could go on but the paper is focused on ocean records. Note that the models used in this paper do include land surface processes, but only 2-3 models allow interactive vegetation (shown in Table 2).

We chose to remain focused on ocean records in this manuscript. The comparison with continental records, with a discussion on sea ice, can also be found in the companion paper on the PMIP4 LIG experiments, by Otto-Bliesner et al.

Comment 2, Page 10: It is now pretty widely accepted that Greenland gave up at least 2 meters of sea level equivalent during MIS 5e (LIG). Dorthe Dahl-Jensen supports this now which is significant! So it would seem to make much more sense that CMIP6 should use the best current configuration of a 5e Greenland Ice sheet. I suggest for clarity that the authors here include an explanation why CMIP6 is not using smaller ice sheets. Lots of examples like The Cryosphere Discuss, https://doi.org/10.5194/tc-2018-225; Stone et al, 2013 C-P; Helsen et al. 2013 also published in CP.

This is now summarised in section 2.4, to which we will add the following lines:

"Both the Greenland and Antarctica ice sheets are known to have shrunk during the interglacial, with different timings, and therefore taking PI characteristics for the *lig127k* protocol is an approximation, in particular for the Antarctic ice sheet which was possibly smaller than PI at that time (Otto-Bliesner et al., 2017). The Greenland ice sheet likely reached a minimum at around 120 ky BP and was probably still close to its PI size at 127ka BP. Given the dating uncertainties and the difficulty for models to include the largest changes in ice sheets for 127 ka BP, i.e. changes in West Antarctica, the choice of the PMIIP4 working group on interglacials was to use the PI ice sheets as boundary conditions for the Tier 1 PMIP4-CMIP6 experiments presented here, and to foster sensitivity experiments to ice sheet characteristics at a later stage. In terms of the Greenland ice sheet, the approximation is considered as quite good and ideal for starting transient experiments through the whole interglacial."

Comment Figure 2 – Add axes labels to all boxes.  Increase the font on the key, there is plenty of space for that.

This has been done. We now include a description in terms of sea ice area (SIA, sum of (sea-ice concentration x ocean cell areas)) and in terms of sea ice extent (SIE, sum of the ocean cells areas for which sea ice concentration is larger than 0.15). The new figure will be similar to the following one, but updated with the latest available models:

[Figure]

Comment 3 page 14 – I could be wrong but what about propagating the error from the proxies given that dinos are +/_ 12%. Evaluate the proxy error vs the model comparison miss match?

The reviewer is right. The marine core community still needs to resolve proxy-related uncertainties, but this is out of the scope of the present contribution. However, the discrepancies are not so large in the subarctic seas and marginal Arctic seas, notably when comparing dinocysts and IP25 data that are providing complementary information on sea-surface productivity. The problem remains in the

central Arctic Ocean, where the issue is related both to the chronostratigraphy and the interpretation of the proxies. Hence, we are now more cautious and avoid proxy-data quantification for most central Arctic sites. Section 2.1 and conclusions are modified accordingly.

Page 15 – You start here using 1pctCo2 for the first time. Please add something to explain this, like... Idealized 1% per year increase in Atmospheric CO2? Etc etc. Remember that not everyone reading this is a modeler so this term should be defined and add why it's important.

We now introduce this experiment at the end of section 2, in a new section:

"2.5 1pctCO2 CMIP6 protocol

We compare the response to the lig127k forcings to idealised forcings for future climate. We have chosen to use the 1pctCO2 simulation from the CMIP6 DECK (Diagnostic, Evaluation and Characterization of Klima, Eyring et al., 2016). These simulations start from the PI (piControl) experiment and the atmospheric CO2 concentration is gradually increased by 1% per year for at least 150 years, i.e. 10 years after atmospheric CO2 quadrupling."

Page 16 – consider this additional important point. Low sedimentation rates in the Arctic Ocean also means that the proxy resolution from the cores you are using are low enough to be missing 1000 yr intervals of no summer sea ice etc. These limitations may also complicate or explain the mixed messages from the 12 models. You should add this to the discussion – a few sentences.

We have taken this comment very seriously in our revised manuscript, where we specifically point to problems in chronology, in particular in the Central Arctic. On the map of Fig. 1, we now use different symbols for records for which the chronology is uncertain and which should be subject to caution. The consequences of these uncertainties are further discussed in the conclusion.

Smaller comments are attached to the text using comment boxes. See the file named "Supplement" for details and picky editorial comments .I suggest this paper be published with minor revisions.

Please also note the supplement to this comment:https://www.clim-past-discuss.net/cp-2019-165/cp-2019-165-RC1-supplement.pdf

Abstract, "21C" is now spelled out.

Line 85: the repetition of "the Northern Hemisphere" will be removed.

Comment 1, page 11 ("Does this also mean late August/early September?"):

We are not sure about the line this comment refers to. In the original manuscript, when we refer to "summer minimum monthly area" (e.g. at the beginning of the second paragraph of section 3.1), we refer to the minimum in the monthly time series of sea ice area, which occurs during summer. This minimum occurs in August or September if the 1850 calendar is used. This can change by 1 month if the 127 ky BP calendar is used, as is shown below for the IPSL model. A short discussion on this "calendar" effect is added in section 3.2.

Comment 2, page 11, about the Bering Sea sea ice not being included in the computation.

Indeed, the Bering Sea is South of 60°N and was therefore not included in our computation of sea ice area shown on Fig.2. In our updated manuscript, for consistency with the SIMIP paper on future sea ice changes, we will consider integrating sea ice cover over the whole northern hemisphere. We will therefore include the Bering Sea sea ice to be more consistent with the maps on Figures 3,4, 6 and 7.

Comment 4, page 11: "CO2 was between 340 and 370 during this interval afterall".

The first paragraph of section 3.1 on sea ice now reads:

"For the present-day we have satellite and in-situ observations with which to evaluate the models. The use of present-day sea ice data implies that we might expect the simulated PI sea ice to be generally somewhat larger than the observed mean. Indeed the atmospheric $CO_2$ levels for the years for which we chose the observation data set (1982 to 2001) were between 340 and 370 ppm, to be compared to the PI level of 280 ppm. Figure 2 shows the mean seasonal cycle of the Arctic sea-ice extent simulated for the PI and LIG alongside the observed Arctic sea-ice extent."

Comment on line 213:

This has been corrected ("To" has been changed to "We".

Comments on Figures 2, 8 and 9:

The axes will be better labelled, as shown on the figure above for Figure 2.

---

## Author Comment (AC3) · 15 May 2020

This response is supplied as a supplementary material.

Please also note the supplement to this comment:
https://www.clim-past-discuss.net/cp-2019-165/cp-2019-165-AC3-supplement.pdf
* * *
[Figure]

Reviewer #2

Kageyama et al. present the results of CMIP6-PMIP4 LIG simulations from 12 models and analyse them in terms of Arctic sea ice changes. They also present a new compilation of LIG sea-ice proxy data which they compare the model results with. While the discrepancies between simulations and proxy data, as well as within proxy data and within simulations, prevent any unambiguous identification of LIG sea-ice changes, the author provide valuable insights into the parameters that may influence sea-ice dynamics through their analysis of inter-model differences.

I find the manuscript well-structured and written in a concise and convincing way, and I only have minor concerns about how the proxy reconstructions were transferred into values of sea-ice concentration and duration (as described below). I thus recommend this manuscript for publication with minor revisions.

We thank the reviewer for this careful review.

—

Major comments about proxy data (mostly section 2.2):

I really like the author's cautious approach to provide common and clear definitions, based on sea-ice cover duration and sea-ice cover concentration, of ice-free / seasonal/ perennial sea ice that facilitate data-data and model-data comparison. However, it is not always clear for me how such values have been obtained for the proxy data:

- For dinocysts:  the explanation is very clear, but I miss the info on how the min and max values have been obtained (are those the min and max values of the 5 (?) best analogues? The minimum and maximum monthly sea-ice cover durations? The range of variability within the LIG time slice? Other?).

The minimum and maximum values are given according to the range of estimates within the LIG time slice. This is added in section 2.2. in which the last sentence in the paragraph about dinocyst data now reads:

"The error of prediction for sea-ice concentration is ±12% and that of sea-ice cover duration through the year is ±1.5 months/yr. Such values are very close to the interannual variability in areas occupied by seasonnal sea-ice cover (cf. de Vernal et al., 2013b)."

- For other proxies:

> I understand the authors attributed values of 0.15 and 0.95 for min and max sea-ice concentration at sites where sea ice was interpreted to be perennial, but I miss the info on how those values were defined for other sea-ice categories (or what are the sea-ice states corresponding to the 3 other min-max SIC combinations: 0.3-0.95, 0.3-0.6 and0.1-0.3).

> The rationale for the attribution of min and max sea-ice cover durations is also not clear to me (in section 3.3 the authors mention they "define perennial sea ice to have at least 9 months of coverage", but I am confused because sites with min-max sea-ice concentrations of 0.15-0.95 have either min-max sea-ice cover durations of 9-12 mth/yr for IP25 or 3-11 mth/yr for faunas).

- Regarding the sites with PIP25-based interpretations, have the attributed min-max range of sea-ice concentration values been compared to some sea-ice concentration quantifications based on the calibrations recently proposed (Xiao et al., 2015, http://dx.doi.org/10.1016/j.gca.2015.01.029; Smik et al., 2016,

**Fig. 1.**

**Supplement:**

Reviewer #2

Kageyama et al. present the results of CMIP6-PMIP4 LIG simulations from 12 models and analyse them in terms of Arctic sea ice changes. They also present a new compilation of LIG sea-ice proxy data which they compare the model results with. While the discrepancies between simulations and proxy data, as well as within proxy data and within simulations, prevent any unambiguous identification of LIG sea-ice changes, the author provide valuable insights into the parameters that may influence sea-ice dynamics through their analysis of inter-model differences.

I find the manuscript well-structured and written in a concise and convincing way, and I only have minor concerns about how the proxy reconstructions were transferred into values of sea-ice concentration and duration (as described below). I thus recommend this manuscript for publication with minor revisions.

We thank the reviewer for this careful review.

—

Major comments about proxy data (mostly section 2.2):

I really like the author's cautious approach to provide common and clear definitions, based on sea-ice cover duration and sea-ice cover concentration, of ice-free / seasonal/ perennial sea ice that facilitate data-data and model-data comparison. However, it is not always clear for me how such values have been obtained for the proxy data:

- For dinocysts: the explanation is very clear, but I miss the info on how the min and max values have been obtained (are those the min and max values of the 5 (?) best analogues? The minimum and maximum monthly sea-ice cover durations? The range of variability within the LIG time slice? Other?).

The minimum and maximum values are given according to the range of estimates within the LIG time slice. This is added in section 2.2. in which the last sentence in the paragraph about dinocyst data now reads:

"The error of prediction for sea-ice concentration is ±12% and that of sea-ice cover duration through the year is ±1.5 months/yr. Such values are very close to the interannual variability in areas occupied by seasonnal sea-ice cover (cf. de Vernal et al., 2013b)."

- For other proxies:

> I understand the authors attributed values of 0.15 and 0.95 for min and max sea-ice concentration at sites where sea ice was interpreted to be perennial, but I miss the info on how those values were defined for other sea-ice categories (or what are the sea-ice states corresponding to the 3 other min-max SIC combinations: 0.3-0.95, 0.3-0.6 and0.1-0.3).

> The rationale for the attribution of min and max sea-ice cover durations is also not clear to me (in section 3.3 the authors mention they "define perennial sea ice to have at least 9 months of coverage", but I am confused because sites with min-max sea-ice concentrations of 0.15-0.95 have either min-max sea-ice cover durations of 9-12 mth/yr for IP25 or 3-11 mth/yr for faunas).

- Regarding the sites with PIP25-based interpretations, have the attributed min-max range of sea-ice concentration values been compared to some sea-ice concentration quantifications based on the calibrations recently proposed (Xiao et al., 2015, http://dx.doi.org/10.1016/j.gca.2015.01.029; Smik et al., 2016,

http://dx.doi.org/10.1016/j.orggeochem.2015.12.007) to see if both methods yield rather similar results?

We have updated Table 1 to respond to this point and considered it is not possible to give quantitative durations of sea ice cover for the Central Arctic cores. These are also the cores with the most uncertain chronology, and this information has also been added to the table, as well as a qualitative description of the sea ice state. The data-model comparison has been updated accordingly.

The new table 1 is as follows:

[revised manuscript text omitted]

- Could it be specified in Table 1 whether it is IP25 and/or PIP25?

The reconstructions are based on both indicators, and this is now specified in Table 1.

—

Minor comments:

-  At my first reading (but not the following), it was not always clear whether it was referred to sea-ice cover duration, concentration or simply sea-ice cover.  Maybe SIC, SICc and SICd abbreviations (or something similar) could be used to help with this?

The reviewer is right, this could be confusing. We use SIC for sea ice concentration as now specified at the end of section 2.2 and in table 1 in the revised manuscript. As explained in our response to reviewer 1, we have also better introduced the terms sea ice area (SIA) and sea ice extent (SIE), which are used in the literature on sea ice. We also define SICd50 as the duration, in months, computed from monthly data, of the period during which SIC > 0.50.

- L18: what is 21C?

21$^{st}$ century, we now spell it out.

- L69: "PI" abbreviation used as "pre-industrial" but defined as such only L133.

Abbreviations has been checked for consistency throughout the manuscript. Our apologies for these inconsistencies. This sentence now reads: "However, a quick assessment of the sea ice simulated in the reference state, i.e. the pre-industrial control experiment (referred to piControl in the CMIP6 terminology, and PI in this manuscript) was necessary."

- Table 1: maybe it would be clearer to specify "duration" in the "sea-ice cover" column (or cf. my first minor comment), as well as "per year" for the unit.

This is now corrected: we refer to "# months during the year with sea ice cover > 50%"

- L90: The error of prediction for sea-ice cover concentration is indicated, but not that for sea-ice duration.

This is now added, cf. our response to the first major point.

- Table 2: some info missing for CESM2 boundary conditions and LIG simulation length, LOVECLIM1.2 physical core components and LIG simulation length, and NESM3 boundary conditions and LIG simulation length.

The following information will be added:

- For CESM2:

       Aerosols: interactive dust

       Spinup (after PI ssinup): 325 years

       Simulation length: 700 years

- For LOVECLIM1.2: the atmosphere component is ECBilt, the ocean and sea ice component is CLIO, the land component is VECODE, the spin-up is 3000 yrs long and the production length is 1000 years.

- For NESM3: interactive vegetation ; the aerosols are prescribed to the pre-industrial values, the ice sheets are prescribed to modern values, the spin up is 500 years-long; the production run is 100 years-long.

- L134: GHG abbreviation not defined

The definition has been added. The sentence is now: "The prescribed LIG (*lig127k*) protocol differs from the CMIP6 Pre-industrial (PI) simulation protocol in astronomical parameters and the atmospheric greenhouse gases concentrations (GHG). "

- L156: should the reference to Figure 3 here be to Table 4 instead, as Figure 3 is referred to 4 lines after when talking about the "The detail of the geographical distribution of sea ice"?

We have included the reference to Table 4 at the end of this sentence (and removed it from the end of the following sentence to avoid repetition)

- L175-177: maybe it would be clearer to mention that the reduction is "between the PI and LIG" in the first sentence rather than/in addition to in the second sentence (as done in the conclusion).

This sentence has been rewritten as: "Thus, compared to the PI results, there is a reduction of 49% in the MMM minimum (summer) monthly SIA in the LIG results, but almost no change for the winter monthly MMM SIA."

- L178: 12 rather than 13 models?

The number of models will be updated (there will be one more model: CNRM-CM6-1).

- L178-181: maybe specify that the third model is NESM3 and refer to the section above regarding the reason why it does not realistically capture the PI Arctic?

Taking both SIA and SIE into account, this paragraph has been updated to:

There is a large amount of inter-model variability for the LIG SIA and SIE during the summer (Figure 4 and Table 4). Out of the sixteen models, one model, HadGEM3, shows a LIG Arctic Ocean free of sea ice in summer, i.e. with an SIE lower than 1 million km2. CESM2 and NESM3 show low SIA values (slightly above 2 mill. km2) in summer for the LIG simulation but their minimum SIE values are around 4 mill. km2. Both HadGEM3 and CESM2 realistically capture the PI Arctic sea ice seasonal cycle. On the other hand, NESM3 overestimates winter ice and the amplitude of the seasonal cycle in SIA and SIE, while simulating realistic PI values for both SIA and SIE (Cao et al., 2018). This seasonal cycle is amplified in the LIG simulation, with an increase in SIA and SIE in winter and a decrease in summer, following the insolation forcing. Hence, the difference in the response of these models to LIG forcing in terms of sea ice does not appear to only depend on differences in PI sea ice representation.

- L188-191 and Figures 6 and 7: "the reconstructed values, classified into 3 categories: perennial cover (9 to 12 months), seasonal cover (3 to 9 months), ice free state (0 to3 months)" –> how were the reconstructions based on the same proxy with e.g. (from Table 1) 3 to 12 months/year classified? Does "ambiguous interpretations" (here but I also mean in general in the MS) refer to those from the same core/area and based on different proxies (which is what I understand) or does it also refer to reconstructions from the same core and same proxy? If so, maybe it would be worth clearly mentioning it too, as it also plays a role in the difficulty to compare model and data (and highlight the proxy limitations from this other perspective) and in model-data discrepancies.

We agree with the reviewer that this was not clear. In the new version of the manuscript, this is described in more detail and Table 1 has been be updated accordingly.

- L191: the first "to" may be removed in "it is not possible to for any one model to match"

Ok, corrected.

- L192: maybe something like "comparison between the PI and LIG model *results*and PI and LIG sea ice *proxy* data" would be clearer

The sentence has been corrected as follows: "the comparison between the PI and LIG results and PI and LIG sea ice reconstructions as a function of the latitude of the LIG data sites is remarkably similar for each individual model (Figure 5)"

- L209-210: I understand that the authors do not want to solve this here, and I think it is not necessary as the focus of this paper is on the models. That said, given the proxy and model dataset presented here, and the authors being one of the world experts on these proxies, I have to admit that I was kind of expecting / hoping for this initially...:-)

=> our apologies for disappointing the reviewer... but we preferred to be honest here and point to remaining work which has to be done.

- L213: "we" instead of "to"

=> this is corrected.

- L224: SW abbreviation not defined

=> SW stand for short wave, indicated just before in the text.

- L276: no need to redefine LIG abbr.

=> ok, we have removed the abbreviation.

- L281: I would maybe rather say "These southern sea ice records are (or "correspond to" or equivalent) quantitative estimates based on dinoflagellate cysts (dinocysts)" to avoid confusion.

=> Fine, this has been changed.

- L288-289: there has been a shortcut, "periods" does not refer to anything here.

=> The sentence is now: "the comparison between the PI and LIG results and PI and LIG sea ice reconstructions as a function of the latitude of the LIG data sites is remarkably similar for each individual model (Figure 5)".

- L299: 12 models + no need to redefine MMM abbr.

=> We redefine the abbreviations in the conclusion in case readers go to them directly. As stated above the number of models will be updated.

- L305-306: needs to be rephrased

The sentence has been rephrased to "In general, the models that fail to realistically represent the numbers of months per year of sea ice cover in the PI also provide unlikely LIG results."

- Figure 1: the colour code for the cores is missing + why are there only some cores labelled? If this is a matter of space, numbers could be used in Table 1 and Figure 1.

=>The figures has been completed. The colour code corresponds to the sea ice proxies as used in other figures throughout the manuscript. Numbers have been added for the sites which were not labelled. The new map is as follows.

[Figure]

| Core name | Map number |
|---|---|
| Oden96/12-1pc | 6 |
| PS2200-5 - Stein et al. (2017) | 8 |
| PS2200-5 - Cronin et al. (2010) | 8 |
| PS51/38-3 | 5 |
| GreenICE | 7 |
| PS92/039-2 | 10 |
| PS2138-1 | 9 |
| PS2138-1 | 9 |
| PS2757-8 | 4 |
| HLY0503-8JPC | 3 |
| NP26-32 | 1 |
| PS93/006-1 | 11 |
| NP26-5 | 2 |
| M23455-3 | 12 |
| M23352 | 13 |
| PS1247 | 14 |
| M23323 | 15 |
| M23071 | 16 |
| MD95-2014 | 17 |
| MD95-2015 | 18 |
| HU90-013-13P | 19 |
| MD95-2004 | 20 |
| HU91-045-91 | 21 |
| IODP1304 | 22 |
| IODP1302/1303 | 23 |
| MD03-2692 | 24 |
| MD95-2042 | 25 |

---

## Author Comment (AC4) · 15 May 2020

Response to Agata De Boer

This paper makes a very important contribution to the quest to understand Arctic Sea-Ice during the LIG, contains interesting diagnostics, a great model-data comparison, and I enjoyed reading it. Because this topic is so important, I'd like to request a few extra figures that may help us understand the models and the LIG Arctic sea ice better. Is there any chance you can produce (perhaps in the supplement?) any of the following variables for the various models for the pre-industrial and LIG? In order of preference:

- Sea-ice drifts or surface currents: Was the surface circulation at the LIG different from today and what are the models showing for the PI? Do they have a realistic PI circulation?

- Winds or windstress over the ocean: This will drive sea-ice export as well as influence sea-ice distributions. It would be interesting to see if there are any consistent changes between the PI and LIG that could be useful in interpreting the data or model differences.

- The barotropic streamfunction or upper layer circulation if different from the surface: Are the models getting the Arctic gyres right and do these gyres systematically change at the LIG? This may also indicate what may happen in the Atlantic layer which is thought to affect sea-ice, at least in the Barents Sea.

- SLP patterns: This should be a straight forward plot indication atmospheric circulation differences between the different models and between the PI and LIG

We thank Agata De Boer for her comments. We will try and add a simple figure with the SLP patterns, since this is the variable which is available for the largest number of models.

A point I'd like to challenge is that models produce a similar response to future $CO_2$ warming and the LIG forcing (Figure 10). If one takes away the one outlier (INMCM4-8) then the correlation breaks down. The response to 1pctCO2 is much stronger in some models than others model, while the response to LIG forcing is similar in the models.

We agree that the relationship is not very strong on the figure of the manuscript. We will reconsider this result in the light of the results made available for additional models.

Figure 5 is very interesting and I keep coming back to it but find it very difficult to dissect what the different models are showing because the symbols overlap to much or are hidden behind the lines.

The data-model comparison figure will be redesigned to account for changes in the interpretation of the data. We will also try to make it clearer.

Thanks for a great paper and I am hoping you will have he means to make some of these clarifying figures. Agatha de Boer

Thank you. This is also meant to be a paper that gives the readers the motivation to carry more analyses of the PMIIP4 data!

---

## Author Response (AR2)

Dear Editor,

Thank you for suggestions for clarifications and improvements. We are providing responses to these below. Your comments are copied in black and answers provided in blue and italics.

I unfortunately cannot create a pdf showing the differences between the last two versions of the manuscript from where I am now, but there are few changes:

- the "pre-industrial" occurrences that have been harmonised (except in references),

- the Data availability section,

- the typo on line 365,

- the caption of Figure 3.

I will need some additional time to prepare the data to be distributed with the paper.

Thank you for your understanding.

Best regards

Masa Kageyama, on behalf of the authors.

- - - - - - - - - - - - - - - - - - - - - - - - - - - - - - - - - - - - - - - - - - - - - - - - - - - - - - - - - - - -

I will be happy to accept your paper for final publication, granted some minor corrections, mostly in the data availability section.

Could you please copy in the data availability section the website address (https://www.esrl.noaa.gov/psd/) where the NOAA_OI_v2 data set is available. Moreover, would it be possible to give the address pointing directly to the dataset and not to the general website.

*Both the data site describing the data set and the address pointing the the data set itself have been added in the "Data availability" section.*

In figure 3, could you please give a reference for 'The magenta contour shows the 0.15 isocontour of the observations averaged over years 1982–2001' as well as the location of the data themselves (doi or website).

The caption of figure 3 has been modified to give the precision about the "observations", the reference and points to the {\it Data availability} section for the source website.

Moreover, CP is not in favor of papers indicating that 'something should happen sometimes' because this will remain like that unless some corrigendum is published. Therefore, statements like 'will be shortly available','the majority of the model simulations' are not acceptable as such. They should be more precise! Where will be available the simulations that are not 'the majority'?

We will make the very data shown on the figures available on a dedicated web site, or as supplementary material if its size is smaller than 50 Mb. We have therefore re-written this part of the "Data Availability" section as follows:

"The majority of the model simulations used in this study are available or will shortly be available on the Earth System Grid Federation (\url{https://esgf-node.llnl.gov/}), the data repository for CMIP6 simulations. The results of analyses from this original data, shown on Figures 2 to 12, will be referred to with a doi and accessible from the IPSL website upon acception of the manuscript."

We will need additional time to prepare this data in a user-friendly format. Can this be accommodated for?

I also identified three spellings for preindustrial, pre-industrial or Pre-industrial. Could this be checked.

We have checked and all occurrences should now be "pre-industrial", as in Eyring et al., 2017's overview paper on CMIP6 simulations.

Non-public comments to the Author:
This is a very specific question. On line 365, what do you mean with 'among the 16th' ? Is it an accidental cut/paste or does it really refer to the sentence. Apologize for this naive question.

This was actually a typo. It should have been « among the 16 models ». Apologies for the confusion it created.

---

## Author Response (AR3)

Dear Editor

Here is a new version of the manuscript, updated with supplementary material ---the data shown in the Figures, for readers to be able to reproduce them. The "data availability" section has been updated accordingly. It specifically refers to the companion paper, by Bette Otto-Bliesner.

Many thanks for your patience

Best regards

Masa Kageyama, on behalf of all co-authors